# U-Surf: a global 1km spatially continuous urban surface property dataset for kilometer-scale urban-resolving Earth system modeling

Yifan Cheng[1], Lei Zhao[1,2,3], TC Chakraborty[4], Keith Oleson[5], Matthias Demuzere[6], Xiaoping Liu[7],
Yangzi Che[7], Weilin Liao[7], Yuyu Zhou[8], Xinchang "Cathy" Li[1]

[1] Department of Civil and Environmental Engineering, University of Illinois Urbana-Champaign, Urbana, IL, USA
[2] Institute for Sustainability, Energy, and Environment (iSEE), University of Illinois Urbana-Champaign, Urbana, IL, USA
[3] National Center for Supercomputing Applications, University of Illinois Urbana-Champaign, Urbana, IL, USA
[4] Atmospheric, Climate, and Earth Sciences Division, Pacific Northwest National Laboratory, Richland, WA, USA
[5] Climate and Global Dynamics Laboratory, NSF National Center for Atmospheric Research, Boulder, Colorado, USA
[6] B-Kode VOF, Ghent, Belgium
[7] Guangdong Key Laboratory for Urbanization and Geo-simulation, School of Geography and Planning, Sun Yat-sen University, Guangzhou 510275, China
[8] Department of Geography, The University of Hong Kong, 999077, Hong Kong, China

*Correspondence to*: Lei Zhao (leizhao@illinois.edu), TC Chakraborty (tc.chakraborty@pnnl.gov), and Weilin Liao (liaoweilin@mail.sysu.edu.cn)

**Abstract.** High-resolution urban climate modeling has faced substantial challenges due to the absence of a globally consistent, spatially continuous, and accurate dataset to represent the spatial heterogeneity of urban surfaces and their biophysical properties. This deficiency has long obstructed the development of urban-resolving Earth System Models (ESMs) and ultra-high-resolution urban climate modeling, particularly relevant for large scales. Here, we present U-Surf, a first-of-its-kind 1km-resolution present-day (circa-2020) global continuous urban surface parameter dataset. Using the urban canopy model (UCM) in the Community Earth System Model as a base model for satisfying dataset requirements, U-Surf leverages the latest advances in remote sensing, machine learning, and cloud computing to provide the most relevant urban surface biophysical parameters, including radiative, morphological, and thermal properties, for UCMs at the facet- and canopy-level. Generated using a systematically unified workflow, U-Surf ensures internal consistency among key parameters, making it the first globally coherent urban canopy surface dataset. U-Surf significantly improves the representation of the urban land heterogeneity both within and across cities globally, provides essential, high-fidelity surface biophysical constraints to urban-resolving ESMs, enables detailed city-to-city comparisons across the globe, and supports next-generation kilometer-resolution Earth system modeling across scales. U-Surf parameters can be easily converted or adapted to various types of UCMs, such as those embedded in weather and regional climate models, as well as air quality models. The fundamental urban surface constraints provided by U-Surf can also be used as features for machine learning models and can have other broad-scale applications for socioeconomic, public health, and urban planning contexts. We expect U-Surf to promote the research frontier on urban systems science, climate-sensitive urban design, and coupled human-Earth systems in the future. The dataset is publicly available at https://doi.org/10.5281/zenodo.11247598 (Cheng et al., 2024).

**1 Introduction**

Urban areas are global hotspots of climate hazards (Intergovernmental Panel On Climate Change, 2023; Robinson et al., 2021; Tabari, 2020; van der Wiel and Bintanja, 2021; Zhao et al., 2021), exposure (Chen et al., 2023; Li et al.,
2019; Yang et al., 2023), and vulnerability (Ajjur and Al-Ghamdi, 2021; Lobo et al., 2023), due to the uniqueness of local urban climates (Baklanov et al., 2018; Cao et al., 2016; Chakraborty et al., 2023; Li and Bou-Zeid, 2013; Zhan et al., 2023, p.201; Zhao et al., 2014, 2018), concentration of population, infrastructure, and capital assets (Gao and Bukovsky, 2023; Masson et al., 2020; Shu et al., 2023), and mixture of diverse communities and socio-ethnic groups (Islam and Winkel, 2017; Kim et al., 2021). With an additional 2.5 billion people projected to reside in urban areas
by 2050 (United Nations, 2018), these climate-driven risks are expected to be exacerbated in future warmer climates (Intergovernmental Panel On Climate Change, 2014). This inevitable urbanization coupled with climate change will expose cities and their residents to greater risks across the world (Feng et al., 2021; Gao and Bukovsky, 2023; Scheuer et al., 2017; Sjöstrand, 2022), but also presents a historic and time-sensitive opportunity to mitigate and adapt to negative climate impacts (Krayenhoff et al., 2018; Zhao, 2018; Zhao et al., 2017). To address this grand challenge, it
is urgent to better understand urbanization and its complex two-way interactions with climate across spatiotemporal scales. Achieving this goal, however, requires advanced data and tools that realistically resolve urban land in models such as mesoscale weather models, Earth system models (ESMs), and Earth System Digital Twins (Li et al., 2023a), both to better understand cities and their impacts and for planning effective climate adaptation and mitigation strategies (Krayenhoff et al., 2021).

In light of the increasingly recognized importance of urban climatic impacts, substantial efforts on representing urban landscapes in local-to-regional climate models have been reported in the past decade, including improved urban-scale process representations (Chen et al., 2011; Conigliaro et al., 2021; Jongen et al., 2024; Langendijk et al., 2024; Lipson et al., 2024) and surface input datasets (Ching et al., 2018; He et al., 2023; Qi et al., 2024; Sun et al., 2021). Urban
representation in global-scale models, however, is significantly lagging. This is because an urban canopy model (UCM) is largely missing in most state-of-the-art ESMs or global climate models (Hertwig et al., 2021; Zhao et al., 2021; Zheng et al., 2021). This omission will become an even more critical issue in the future, as next-generation ESMs are expected to run at kilometric scales (Schär et al., 2020, 2021; Wang et al., 2022; Yuan et al., 2023), at which resolving urban areas, their unique biophysical properties, changes over time, and interactions with broader-scale systems will
inevitably be required (Chakraborty and Qian, 2024; Grimmond et al., 2011; Sharma et al., 2021). One primary roadblock that has prevented the development of "urban-resolving" ESMs and accurate global urban climate modeling for decades is the lack of globally consistent estimates of urban surface properties, which are critical model inputs, especially at fine resolutions.

Currently, there is no global and spatially continuous urban dataset that can provide all relevant biophysical parameters for UCMs that can be used in state-of-the-art ESMs across scales (Masson et al., 2020). Unlike local- and regional-scale studies using mesoscale UCMs, for which the urban surface parameters usually rely on either simple look-up tables or user-supplied locally-defined physical description of the study area, common UCMs embedded in ESMs

need complete, fine-resolution, globally and internally consistent, and spatially explicit urban surface parameters. These parameters are required at the facet and canopy level, to be in line with the structural assumptions of the underlying model, and therefore, are dramatically challenging to produce at the global scale.

An urban surface data created by Jackson et al., 2010 (hereafter referred to as J2010) is, to our knowledge, the only available global dataset to date that can provide all the required UCM parameters for Earth system modeling in a globally consistent manner. It was developed by synthesizing population density estimates, satellite data, existing literature, building codes, and municipal documentation. This dataset and its updated version (Oleson and Feddema, 2020) serve as the default urban surface property input for the Community Earth System Model version 2 (CESM2) (Danabasoglu et al., 2020) and Energy Exascale Earth System Model (E3SM) (Golaz et al., 2022). Compiled at a time when fine-resolution geospatial data were very scarce, J2010 is coarse-grained, spatially discontinuous, and somewhat outdated (valid for circa-2000), and hence poorly constrains the spatial heterogeneity of urban properties within cities and across the world. J2010 clusters the global urban areas into 33 distinct regions of similar climates, socio-economic characteristics, and architectural practices (Figure S1), with properties defined within each region for up to four urban density classes: low density (LD), medium density (MD), high density (HD), and tall building district (TBD). These density classes are classified based on morphological features (including building height, pervious areal fraction, canyon height-to-width ratio, and typical building type) and population density. The dataset then prescribes uniform surface properties to each density type within a region. These simplistic, coarse-grained, and region-based urban property constraints impede its application in resolving the true heterogeneity of cities and their interactions with background climate, especially relevant for high-resolution urban climate modeling.

Recent development of the Local Climate Zone (LCZ) framework (Stewart and Oke, 2012) provides another potential means to supply spatially explicit urban parameters to regional and global models. LCZ standardizes a common descriptive methodology to classify land surfaces into 10 built and 7 natural land cover types, each associated with some prescribed ranges of values for a subset of (mostly morphological) parameters. Compared to the widely used conventional land cover maps or density classes, LCZs are a step forward for representing some additional spatial heterogeneity of urban landscapes (Demuzere et al., 2022a). Many high-resolution regional (Demuzere et al., 2021; Huang et al., 2021; Qi et al., 2024) and global (Demuzere et al., 2022a) LCZ raster maps have been produced in recent years, greatly advancing the description of urban typologies at large scales in a "universal" way. However, a critical gap that remains is how to determine the urban canopy parameters based on the LCZ raster maps. A common approach currently relies on referring to the predefined value ranges from the original LCZ typology (Demuzere et al., 2022b; Stewart and Oke, 2012; Sun et al., 2021), which essentially remains a look-up-table method with large degrees of freedom. Similar to J2010, in the LCZ framework, the inherent assumption of uniformity within each zone, i.e. cities located in different countries and diverse climate regimes being assigned the same set of parameters if classified as one LCZ type, oversimplifies the complexity and heterogeneity of urban surfaces. In addition, LCZs, by their very nature, largely describe typologies of urban morphology, but other characteristics such as radiative properties and construction materials are less well defined and subject to large uncertainties (Hidalgo et al., 2019; Masson et al.,

2020). More importantly, these properties can be frequently decoupled from that morphology, meaning the complete set of parameters used as model inputs are not internally consistent.

To address this long-standing urban representation challenge at large scales and to facilitate next-generation kilometer-scale (k-scale) urban-resolving Earth system modeling, we develop a first-of-its-kind global high-resolution (1 km) urban surface dataset, namely U-Surf, to support urban climate modeling across scales. The development of U-Surf is enabled by latest advances in high-resolution remote sensing measurements from recent satellite missions, new algorithms to derive satellite-derived products, building footprint estimates from global scale image segmentation methods, and advancements in hybrid cloud supercomputing. We use the urban scheme in CESM2's land model (Community Land Model or CLM) as the base model to develop the dataset, as it is one of the very few state-of-the-art ESMs with an urban canopy representation. Nevertheless, the derived parameters in U-Surf can be easily adapted to other mesoscale weather or global climate models such as The Weather Research and Forecasting Model (WRF) and E3SM, the latter using a UCM identical to that in CLM version 4.5. The U-Surf data does not rely on any coarse-graining (clustering), but instead estimates the facet- and canyon-level surface properties in a spatially continuous manner at 1 km resolution. Therefore, the final U-Surf product provides a global, internally consistent and comprehensive set of urban surface input for UCMs, captures the fine-scale spatial heterogeneity both within and across cities, and markedly advances the potential for urban representation in weather and climate models across scales. In addition to its applications in climate modeling, U-Surf could be used directly as input features for machine learning models, and can also be leveraged for other non-climatic modeling exercises, analyses, or applications in the energy, geography, and socioeconomic fields.

This paper is organized as follows. Section 2 details the data sources and methodology employed in developing the dataset. Section 3 presents the spatial distributions of the newly created 1km-resolution dataset, highlighting selected parameters across various scales. Sections 4 and 5 discuss the broad implications of the dataset, the current limitations, and potential future work. Section 6 provides information and links on accessing the dataset in different formats and associated Google Earth Engine (GEE) web application, while section 7 provides concluding remarks.

## 2 Data and methods

### 2.1 Urban representation in CESM2

The current version of the U-Surf dataset is based on the urban parameterization scheme embedded in the CESM2 for two reasons. First, CESM2 is one of the very few state-of-the-art ESMs with a physically based UCM – the Community Land Model Urban (CLMU) (Lawrence et al., 2019; Oleson and Feddema, 2020). CLMU has been evaluated against site observations and satellite measurements across the world with consistently reasonable agreement (Demuzere et al., 2008, 2013, 2014, 2017; Fitria et al., 2019; Li et al., 2024a, b; Lin et al., 2016; Oleson et al., 2008a, b; Zhang et al., 2023a; Zhao et al., 2014, 2021) and has also demonstrated high credibility among various UCMs in the recent Urban-PLUMBER multi-model comparison project (Lipson et al., 2024). Second, the urban

canopy concept that CLMU uses is widely adopted in various UCMs embedded in weather models and RCMs. Therefore, a dataset developed based on this conceptual representation can be easily extended to other UCMs within climate and weather models.

The urban canopy representation used in CLMU and many other UCMs is called an "urban canyon" schema, where the urban landscape at a given location is conceptualized as an infinite "urban canyon" (Figure 1). This canyon hypothesis assumes a geometry of an infinitely long street bordered by two building walls with identical height. An "urban canyon" consists of five facets: building roof, impervious (e.g., roads, parking lots, sidewalks) and pervious (e.g., lawns, street trees, parks) canyon floors, and sunlit and sun-shaded walls (Oleson et al., 2008a). This conceptual
representation reduces the considerable complexity of urban surfaces into a single urban canyon, and yet provides an essential base to represent key urban biogeophysical processes effectively. The UCMs using this approach therefore require sets of properties at both facet- and canopy-level to represent urban landscapes and model their interactions with the lower atmosphere in climate/weather simulations. These properties can be generally grouped into three categories: morphological (e.g., canyon height-to-street width ratio, roof fraction, average building height, and
pervious canyon floor fraction), radiative (e.g., facet-level albedo and emissivity), and thermal (e.g., heat capacity and thermal conductivity) (Figure 1 and Table 1). These surface properties characterize the "urban areas" and are critical for constraining their surface energy budget, and thus the near-surface microclimate, in weather and climate models. More details about the CLMU parameterization scheme and its evolution over the years can be found in Jackson et al. (2010), Lawrence et al. (2019), Li et al. (2024b), Oleson et al. (2008a, b), Oleson and Feddema (2020).

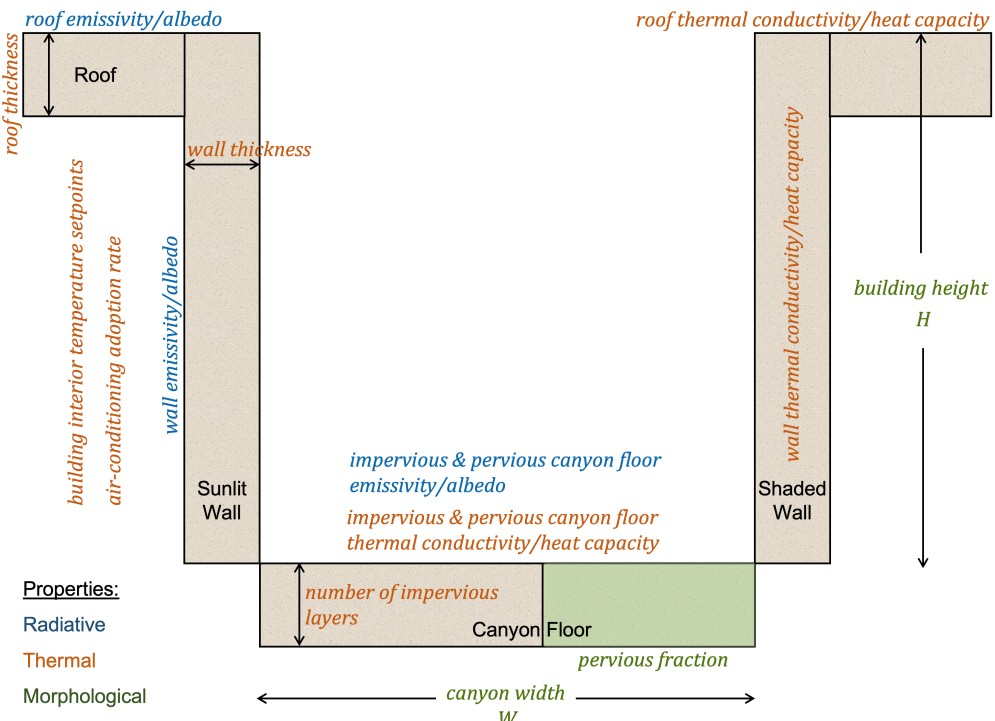

**Figure 1. Conceptual schematic of an urban canyon to represent urban landscapes in CLMU** (adapted from Oleson et al., 2008a). Properties are color-coded: blue for radiative, orange for thermal and green for morphological. Note that roof and wall thickness (despite being related to urban morphology) are considered thermal properties, as they are primarily used as weighting factors to calculate conduction fluxes into and out of canyon surfaces in CLMU (Lawrance et al., 2018; Oleson et al., 2010).

 **Table 1. Data sources and retrieval methods for each urban parameter in U-Surf and the CLMU dataset.**

| Category | Urban Parameters | U-Surf | CLMU |
|---|---|---|---|
| **Radiative** | Roof Emissivity | • Source: 100m ASTER v3 emissivity product (Hulley et al., 2015) and broadband emissivity algorithm (Malakar et al., 2018; Ogawa et al., 2008)<br>• Time span: static, representing 2000-2008<br>• Spatial resolution: 1km | • Source: local building codes, municipal documentation, literature, satellite imageries (Jackson et al., 2010; Oleson and Feddema, 2020)<br>• Time span: 1966-2007<br>• Spatial resolution: Regional-level, density-class-specific |
| | Impervious Canyon Floor Emissivity | | |
| | Pervious Canyon Floor Emissivity | | |
| | Wall Emissivity* | | |
| | Roof Albedo | • Source: 10m Sentinel2 albedo product (Lin et al., 2022) and narrow-to-broadband algorithm (Bonafoni & Sekertekin, 2020)<br>• Time span: 2019-2021<br>• Spatial resolution: 1km | |
| | Impervious Canyon Floor Albedo | | |
| | Pervious Canyon Floor Albedo | | |
| | Wall Albedo* | | |
| **Morphological** | Building Height | • Source: 3D-GloBFP (Che et al., 2024) and 3D building structure (Li et al., 2022)<br>• Time span: 2014-2023 and circa-2015<br>• Spatial resolution: 1km | • Source: local building codes, municipal documentation, literature, satellite imageries (Jackson et al., 2010; Oleson and Feddema, 2020)<br>• Time span: 2000-2007<br>• Spatial resolution: Regional-level, density-class-specific |
| | Canyon Height-to-width Ratio | • Source: infinite canyon street model (Masson et al., 2020)<br>• Time span: 2014-2022<br>• Spatial resolution: 1km | |
| | Roof Fraction | • Source: Microsoft global building footprints (Microsoft, 2022), East Asia building footprints (Shi et al., 2024)<br>• Time span: 2014-2022<br>• Spatial resolution: 1km | |
| | Pervious Canyon Floor Fraction | • Source: 10m ESA Worldcover v200 (Zanaga et al., 2022)<br>• Time span: 2021<br>• Spatial resolution: 1km | |
| | Urban Percentage | • Source: building footprints (Microsoft, 2022; Shi et al., 2024) and ESA (Zanaga et al., 2022)<br>• Time span: 2014-2022<br>• Spatial resolution: 1km | • Source: LandScan global population database (Bright et al., 2005)<br>• Time span: 2004<br>• Spatial resolution: 1km |
| **Thermal** | Air Conditioning Penetration Rate | • Source: global AC penetration rate and constant maximum interior building temperature of 300K (Li et al., 2024b)<br>• Time span: present-day, loosely defined as 2010-2020<br>• Spatial resolution: national and sub-national level | AC penetration rate is not explicitly modeled in CLMU as of CLM5 (Oleson and Feddema, 2020), maximum interior building temperature is varied by regions and density classes. |
| | Maximum Interior Building Temperature | | |
| | Number of Impervious Canyon Layers | • Source: local building codes, municipal documentation, literature, satellite imageries (Jackson et al., 2010; Oleson and Feddema, 2020)<br>• Time span: 1966-2007<br>• Resolution: Regional-level, density-class-specific** | |
| | Roof Thickness | | |
| | Wall Thickness | | |
| | Minimum Interior Building Temperature | | |
| | Roof Thermal Conductivity | | |
| | Impervious Canyon Thermal Conductivity | | |
| | Wall Thermal Conductivity | | |
| | Roof Volumetric Heat Capacity | | |
| | Impervious Canyon Volumetric Heat Capacity | | |
| | Wall Volumetric Heat Capacity | | |

*Wall emissivity and albedo are derived leveraging the remote sensing data and CESM2 default J2010 radiative data which was based on building materials; **Although thermal properties in U-Surf are provided at a 1 km resolution, the values are derived from regional-level and density-class-specific properties from Oleson and Feddema (2020).

## 2.2 Development of the U-Surf urban parameters

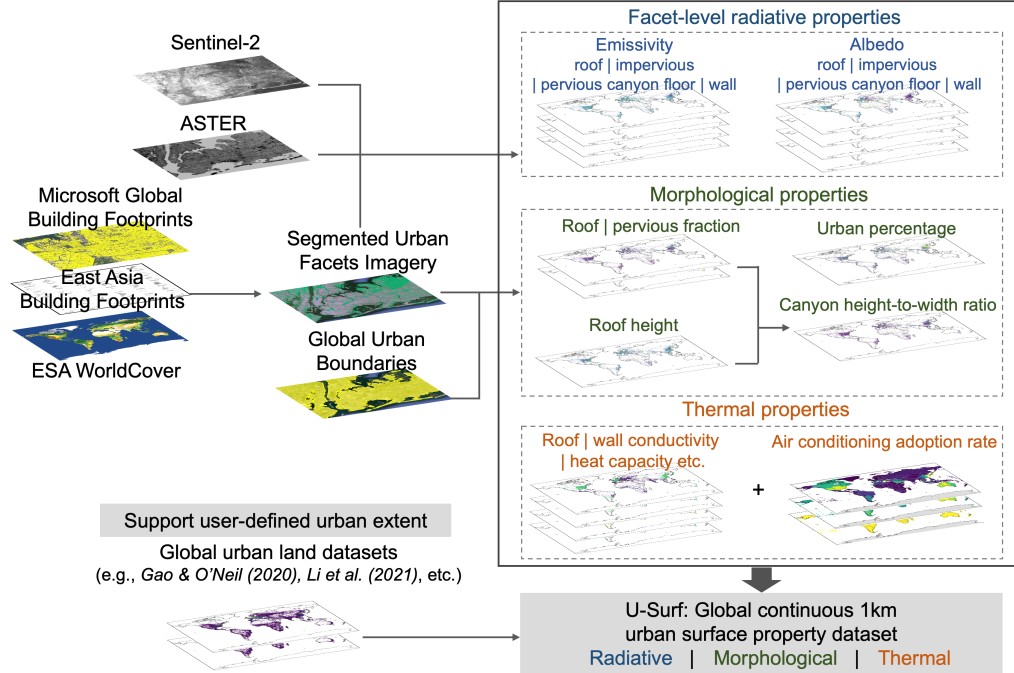

**Figure 2. Overview of data synthesis workflow, including individual data sources and examples of final data product layers.** The satellite images are accessed through and workflow is implemented on Google Earth Engine (Gorelick et al., 2017).

The new urban surface parameter dataset, U-Surf, describes the global urban areas in a spatially continuous manner, providing all the required parameters in three categories (radiative, morphological, and thermal) that are compatible with the urban canyon representation in CLMU and potentially also for other UCMs. We developed a multi-step workflow on the Google Earth Engine platform (Gorelick et al., 2017) that leverages four key categories of data: segmented land use/land cover map, 3D building footprints, high-resolution satellite observations, and thermal properties of construction materials. Utilizing these, we first generated segmented urban imagery, which distinguishes among different facets. Then we integrated this imagery with satellite observations to derive facet-level radiative properties and fractional parameters. From there, we synthesized multiple data sources to construct the 3D urban canyon morphological attributes. Finally, we incorporated existing databases to produce thermal properties (Figure 2).

### 2.2.1 Radiative parameters

To derive facet-level radiative properties, we first needed to identify individual facets such as building roofs, impervious and pervious ground within each 1 km grid. Here in this study, we use the open-source vector-based Microsoft Global Building Footprints dataset (hereafter referred to as MS-BFP; Microsoft, 2022) in conjunction with the East-Asia building footprints (hereafter referred to as EA-BFP) from Shi et al. (2024) and Che et al. (2024) to identify building roofs. The additional East-Asia dataset is necessary due to the insufficient building vectors in the current version of MS-BFP for that region. We then combined this data with the European Space Agency (ESA) WorldCover (Zanaga et al., 2022), a 10 m resolution global land cover product based on Sentinel-1 and 2 data, to

characterize impervious and pervious canyon floors. We choose the ESA WorldCover instead of other available global 10 m land cover products since its 'built-up' class is definitionally consistent with the impervious surfaces in CLMU (Chakraborty et al., 2024). Accordingly, the impervious canyon floor was estimated by subtracting the roof pixels (derived from the MS-BFP and EA-BFP vectors) from areas classified as "built-up" class; whereas pervious surfaces are estimated by aggregating the "tree cover", "shrubland", "grassland", and "bare or sparsely vegetated" areas identified in the ESA WorldCover. This process results in a segmented global urban facet image that serves as the foundation for our subsequent derivation of facet- and canopy-level radiative and morphological parameters.

This facet-segmented image was then applied to the Advanced Spaceborne Thermal Emission and Reflection (ASTER) Global Emissivity Dataset 100 m V003 product (hereafter referred to as ASTER GEDv3; Hulley et al., 2015) and the Sentinel-2 land surface albedo data (Lin et al., 2022) to extract the emissivity and albedo of building roof, and impervious and pervious ground. The static emissivity imagery is composited from clear-sky (cloud free) pixels for all available ASTER data from 2000 to 2008 (Hulley et al., 2015) to represent the emissivity climatology over this period. We use a linear spectral-to-broadband algorithm (Malakar et al., 2018) to estimate the broadband emissivity from ASTER GEDv3 bands (Eq. 1):

$$\varepsilon_b = 0.128 + 0.014\varepsilon_a^{10} + 0.145\varepsilon_a^{11} + 0.241\varepsilon_a^{12} + 0.467\varepsilon_a^{13} + 0.004\varepsilon_a^{14} \tag{1}$$

where $\varepsilon_b$ is the broadband emissivity, $\varepsilon_a^{10}$ to $\varepsilon_a^{14}$ denote the ASTER mean emissivity of bands 10 to 14, respectively, which are the five thermal infrared bands with 90 m resolution.

Note that the 100 m resolution of ASTER GEDv3 could be too coarse for certain small individual facets (e.g., small roof tops, narrow roads between buildings) and thus a potential source of uncertainty. However, given the relatively narrow range of emissivity values (i.e., near blackbody) of typical materials and natural surfaces (Oke et al., 2017), this uncertainty is likely small.

For albedo, we used a 10 m land surface blue-sky albedo product retrieved from Sentinel-2 which covers nearly 2,300 major cities across the globe (Lin et al., 2022). For the rest of the global urban areas, we applied the narrow-to-broadband conversion method (Bonafoni and Sekertekin, 2020) to estimate the 10m-resolution albedo based on Sentinel-2 surface reflectance (Eq. 2). Both the blue-sky albedo product and the narrow-to-broadband calculated albedo are derived using the Sentinel-2 imageries composited from 2019 to 2021. The blue-sky albedo product only includes cloud free images. For the narrow-to-broadband algorithm, we use the Cloud Score+ (CS+) dataset (Pasquarella et al., 2023) to mask out the cloud-contaminated pixels, where pixels with a CS+ quality assessment score below 0.8 were excluded.

$$\alpha = 0.2266\rho_{B2} + 0.1236\rho_{B3} + 0.1573\rho_{B4} + 0.3417\rho_{B8} + 0.1170\rho_{B11} + 0.0338\rho_{B12} \tag{2}$$

where $\alpha$ is the broadband surface albedo, $\rho_{B2}$ to $\rho_{B12}$ represent the surface reflectance for bands B2 to B12 of Sentinel-2 Multispectral Instrument (MSI) respectively. The 10 m resolution Sentinel-2 albedo data provides the fine granularity to differentiate between roof, impervious, and pervious canyon floor.

The derived facet-level emissivity and albedo were then aggregated to the 1km grids using an area-weighted approach:

$$\varepsilon_f^{1km} = \frac{\sum_{i=0}^{99} w_f^i \cdot \varepsilon_i^{100m}}{\sum_{i=0}^{99} w_f^i} \tag{3}$$

$$\alpha_f^{1km} = \frac{\sum_{i=0}^{9999} w_f^i \cdot \alpha_i^{10m}}{\sum_{i=0}^{9999} w_f^i} \tag{4}$$

where $\varepsilon_f^{1km}$ and $\alpha_f^{1km}$ is 1 km emissivity and albedo, respectively, for a certain facet (roof/impervious canyon floor/pervious canyon floor), $w_f^i$ is the area fractions of a certain facet within each 100m or 10m grid cell derived from the 10m segmented imagery. The subscript $f$ stands for each individual facet. For example, when calculating roof emissivity and albedo, $w_f^i$ is the roof fraction within each 100m and 10m grid cell, respectively.

Because satellites mostly sample roofs (canopy tops) and canyon floors, wall emissivity and albedo can hardly be measured from passive satellite remote sensing. To address this issue, we leveraged the CLMU radiative data which was based on building materials. Specifically, for wall emissivity, we assume it shares the same/similar emissivity as its building roof, since the wall surfaces within the same building could either have analogous material compositions in interior layers with roofs or not deviate much in terms of emissivity value given the nature of its narrow range. This will be further discussed in Results and Discussion (Sect. 3.1 and 3.2). For wall albedo, we assume that the ratio of material-based roof albedo to wall albedo in J2010 approximately holds for our data. We then applied this roof-to-wall albedo ratio calculated from J2010 to our new satellite-based roof albedo to derive the wall albedo at each 1 km grid.

## 2.2.2 Morphological parameters

The morphological parameters in the "urban canyon" conceptual model (particularly the fractional parameters) are normally defined with respect to its corresponding "urban" landscape. In U-Surf, we combined the Global Urban Boundaries (GUB, Li et al., 2020) and the ESA WorldCover data to identify and sufficiently preserve "urban" or "built-up" landscapes as much as possible. Developed based on the Global Artificial Impervious Area data (Gong et al., 2020), the GUB dataset provides a collection of physical boundaries of global urban extents. We first overlaid the GUB polygons with the ESA WorldCover map to identify all "urban" surfaces recognized by GUB. For the grids falling outside of the urban boundaries, we applied a 10×10 (i.e., 100m × 100m) window on the ESA WorldCover data and calculated its "built-up" fraction (i.e., the sum of roof and impervious canyon floor fraction) within the window. If the "built-up" fraction is larger than 10%, we define the window as "urban". We chose a threshold of 10%, which is at the lower end of the typical thresholds used in the literature (10% - 30%), to preserve as many "urban" grids as possible. The urban fraction was then calculated based on the proportional areas of roof, impervious and pervious canyon floor, following the GUB-defined thresholds. Although this will likely result in an inclusion of some "sub-urban" landscapes in the U-Surf raw data, users have the flexibility to apply stricter criteria (larger built-up thresholds) to extract "urban" grids according to their own definition. This approach is designed to maximize the retention of grids, ensuring U-Surf's adaptability to various user-defined urban extents (Figure 2; e.g., (Gao and O'Neill, 2020; Li et al., 2021; Zhao et al., 2022; Zhou et al., 2015)). The roof fraction is then defined as the ratio

between roof area and urban horizontal surface area, where roof area is calculated from the building footprints polygons. Consistent with the definition in CLMU, the pervious fraction is defined as the ratio of pervious canyon floor to the sum of impervious and pervious canyon floors.

The building height ($H$) was obtained primarily from the 3D-GloBFP data (Che et al., 2024) and supplemented by another building height dataset by Li et al. (2022) to maximize the spatial coverage. 3D-GloBFP is a global building height data at a building footprint scale recently developed by leveraging a combination of Synthetic Aperture Radar (SAR), optical imagery, terrain, population, nighttime light data primarily covering 2014 to 2023, and XGBoost

machine learning approach. We aggregated the vector-level height to 1 km grids using area-weighted averages. The second global building height data (Li et al., 2022) is a raster map at 1 km spatial resolution that also utilizes radar and optical satellite imagery, along with additional geographical information circa-2015. To comply with the CLMU requirement, we calculated another building height-related parameter: the height at which wind speed in urban canyons is computed. This parameter is simply set at half the building height in the current version of CLMU, providing a

standardized reference point for wind calculations in urban environments.

Canyon height-to-width ratio ($H/W$; i.e., the ratio of building height to canyon width) is another critical morphological parameter that is widely used in most UCMs including CLMU. It is a proxy parameter that captures the structural layout and compactness of the built area. Unlike other parameters that can be directly measured by satellite data, $H/W$

needs to be derived on the basis of model geometry and assumptions. Consistent with the single-layer urban canyon geometry in UCMs, the $H/W$ in this study is estimated using the 2D infinite street canyon model with two recommended primary parameters, building fraction (or plan area density; $\lambda_p$) and wall surface density ($\lambda_w$) (Masson et al., 2020):

$$H/W = \frac{\lambda_w}{2(1-\lambda_p)} \tag{5}$$

where $\lambda_w$ is calculated as the ratio between the surface wall area that is in direct contact with the atmosphere (i.e., external wall surfaces, $A_w$) and the horizontal urban surfaces, as represented in the building footprints (Microsoft, 2022; Che et al., 2024; Shi et al., 2024); and $\lambda_p$ is building fraction (i.e., roof fraction) as described above. The external wall surfaces area is estimated by

$$A_w = N \cdot P_b \cdot H_b \tag{6}$$

where $N$, $P_b$ and $H_b$ are the number of buildings, the average perimeter of buildings, and the height of buildings within each 1km grid, respectively.

### 2.2.3 Thermal parameters

Requirements of thermal parameters are relatively diverse compared to radiative and morphological parameters among various UCMs, depending not only on the UCM's parameterization itself but also on whether and what type of a

building energy model is in place (Reinhart and Cerezo Davila, 2016; Sezer et al., 2023). The thermal parameters required in CLMU include volumetric heat capacity and thermal conductivity of roofs, impervious canyon floors and

walls, thickness of roofs and walls, minimum/maximum building interior temperature, as well as the penetration rate of air conditioning (AC). These parameters are exceptionally challenging to acquire on a large scale, as they cannot be detected by satellite remote sensing. The most feasible way to estimate these parameters by far is still from information about dominant construction materials combined with local surveys and building code, which is largely the approach used in J2010.

However, because of the coarse resolution of previous versions of CLMU, the capability of the J2010 thermal parameters data has not yet been taken full advantage of. Here we adapt the thermal parameters from J2010 raw data to U-Surf, aiming to better leverage its material-based estimates. J2010 compiled a comprehensive look-up table based on the thermal properties of 49 types of construction materials from imagery, construction data and documentations by country (Jackson et al., 2010). This table includes thickness, thermal conductivity and volumetric heat capacity of up to 10 layers for common types of roofs, walls, roads (layers with identical materials are allowed) (Oleson and Feddema, 2020). As these thermal parameters are provided in a look-up table instead of a geospatially explicit format, we need to map the table values to each 1 km grid in U-Surf. In order to do this, we classified 1km U-Surf urban grids into four nominal density classes: TBD (0.016% of the pixels), HD (3.83%), MD (41.98%), LD (54.17 %) (Figure S2) based on the percentiles of canyon height-to-width ratio defined in J2010. We then applied the corresponding thermal parameters from the lookup table to each class to ensure it covers all possible materials used in the 33 regions (Figures S18-S25). Although this is likely the most feasible approach for providing an ESM-compatible global building thermal property dataset at present, we acknowledge its limitation of relying somewhat on coarse-grained regional and density-class values. Once more detailed, spatially explicit global datasets – such as those on building materials or thermal properties – become available, we can readily incorporate their thermal parameters into future releases of U-Surf.

The AC adoption rate ($P_{AC}$) is a new thermal parameter added to the latest version of CLMU/CESM because of the introduction of a new explicit-AC-adoption scheme in the building energy model of CLMU (Li et al., 2024b). Along with this new scheme, Li et al. (2024b) also created a present-day, global, survey-based, and spatially explicit AC adoption rate dataset at country and sub-country level. The AC adoption rate data are created by leveraging U.S. EIA data, literature reports, national surveys, government documentation, the AC units per household data from the International Energy Agency (IEA). To comply with this energy scheme, the maximum building interior temperature is set to a constant value of 300K globally. More details on this new $P_{AC}$ data are discussed in Li et al. (2024b). We incorporated this $P_{AC}$ dataset into our new U-Surf dataset by producing the density-class-weighted averages at 1 km resolution.

All the source data, estimation and/or processing methods, and the comparison with CLMU urban surface data are summarized in Table 1.

**Table 2. Quality control flags for U-Surf dataset.** Note that percentages in the parentheses represent the percentage of grid cells with the corresponding QC flag.

| Category | Radiative | | | | | | | | | | | Morphological | |
| --- | --- | --- | --- | --- | --- | --- | --- | --- | --- | --- | --- | --- | --- |
| | Emissivity | | | | Albedo | | | | Fraction | | | Building height | Canyon height-to-width ratio |
| Variable | Roof | Impervious canyon floor | Pervious canyon floor | Wall | Roof | Impervious canyon floor | Pervious canyon floor | Wall | Roof | Pervious canyon floor | Urban percentage | | |
| 1st digit (Algorithm) | 1 | 1 | 1 | 2 | 1 | 1 | 1 | 2 | 1 | 1 | 1 | 3 | 2 |
| | 1: Processing based on observation products, 2: Processing based on model/assumptions, 3: Regridding of existed products w/o further change | | | | | | | | | | | | |
| 2nd digit (Source) | 0 | 0 | 0 | 0 | 1 | 1 | 1 | 1 | 0 | 0 | 0 | 1 | 1 |
| | 0: Single source, 1: Multiple sources | | | | | | | | | | | | |
| 3rd digit (Gapfill) | 00, 99 | 00, 99 | 00, 99 | 00, 99 | 00, 99 | 00, 99 | 00, 99 | 00, 99 | 00 | 00 | 00 | 00, 99 | 00, 99 |
| | 00: Direct derivation, 99: Gapfilled values | | | | | | | | | | | | |
| QC_flag | 1000 (98.34%), 1099 (1.66%) | 1000 (98.26%), 1099 (1.74%) | 1000 (98.56%), 1099 (1.44%) | 2000 (98.34%), 2099 (1.66%) | 1100 (98.22%), 1199 (1.78%) | 1100 (98.54%), 1199 (1.46%) | 1100 (98.45%), 1199 (1.55%) | 2100 (98.21%), 2199 (1.79%) | 1000 (100.00%) | 1000 (100.00%) | 1000 (100.00%) | 3100 (96.56%), 3199 (3.44%) | 2100 (96.56%), 2199 (3.44%) |

## 2.3 Masking, gap filling and quality control

After estimating all the required parameters as described above, we took several additional steps to ensure the accuracy,
coherence, and transparency of our U-Surf data product, including masking, gap filling, and quality control. First, we
only retain the grids containing all three facets – roofs, impervious and pervious canyon floors in U-Surf, because a
complete "urban canyon" can only be formed when all the three facets are present. This constraint helps make U-Surf
more consistent with the conceptual definition of physical urban land in an UCM and is an improvement over the
J2010 dataset, which used urban density classes from urban form and population estimates (LandScan), leading to
large over- and under-estimations of physical urbanization depending on region (Chakraborty et al., 2024). Second,
we masked out the grids with extremely high canyon height-to-width ratios (>12) yet low building heights (<40 m).
These grids are actually very sparsely built suburban or rural landscapes instead of densely built areas.

The last step is to gap-fill the missing values caused by synthesizing multiple datasets with different spatial coverage.
For example, the emissivity product from ASTER GEDv3 has missing pixels in certain regions due to cloud coverage.
These missing values were gap-filled using a simple approach. We combined two classification data: the Koppen
climate zones (Beck et al., 2018) and the 33 urban regions defined in J2010, both at 1 km-resolution. The average
parameter values for each combined class were then used to fill the missing values of the corresponding parameters.
Note that only a small proportion of grids needs to be gap-filled, accounting for less than 3.5% of the total among all
parameters. To keep the aforementioned data source, processing, and gap-filling information accessible and to make
it easier for users to track changes in future version releases, each parameter comes with an additional quality control
(QC) band using a 4-digit code (Table 2). The first and second digits differentiate algorithms and single/multi source
data, respectively, while the last two digits indicate whether the parameter was directly derived or gap filled. These
QC codes are consistent across the entire dataset and will be updated accordingly in later versions.

## 2.4 Dataset validation

Validating urban surface parameters on the global scale is extremely challenging primarily due to the lack of globally
consistent measurement networks. This challenge is exacerbated by the scarcity of long-term urban observational sites,
especially in diverse urban environments. The inherent variability within urban areas further complicates validation
efforts, as data from one site may not represent the broader urban landscape. U-Surf is composed of extraction of
satellite measurement, satellite-derived products (i.e., land cover data and building footprints), and our own derived
parameters. The satellite measurements and derived products have already been validated and quality-controlled by
their development teams, as summarized in Table 3. U-Surf parameters derived based on these input data sources are
therefore subject to their inherent uncertainties and uncertainty propagation during data synthesis and processing. To
systematically evaluate the accuracy and uncertainty of U-Surf parameters, we first conducted a thematic validation
on the derived morphological parameters at 1 km resolution against the 3D World Settlement Footprint (WSF-3D,
Esch et al., 2022) observational site data and Urban-PLUMBER site metadata. We then further employed Monte Carlo

simulations to quantify the final uncertainties of U-Surf parameters arising from input data errors/uncertainties and their propagation (see Sect. 3.4 for detailed discussion).

## 3 Results and discussion

### 3.1 Global distribution of 1km urban surface property parameters

U-Surf demonstrates significant improvements over the default CLMU parameters. As U-Surf directly provides spatially continuous urban surface parameters without relying on any density class or land use classification, here just for the ease of comparison and illustrative purpose, we separated raw U-Surf pixels into the four urban density classes (TBD, HD, MD, and LD) following their locations defined by J2010 and plotted the distributions of the urban surface parameters in both U-Surf and J2010 data at these locations (Figure 3). The location data defined by J2010/OF2020 at 1 km resolution can be accessed at https://doi.org/10.6084/m9.figshare.28169324.v1. The overall distribution of U-Surf raw data is also shown in the figure. As most of the thermal properties in U-Surf are adapted from J2010, the discussion here will be mainly focused on radiative and morphological parameters and the comparison of thermal properties can be found in Figure S3.

Retrieved from direct remote sensing measurements, the radiative properties exhibit physically more reasonable ranges compared to J2010 data. As described above, urban surface properties in J2010 were estimated on the basis of building materials sampled in a predefined region and then generalized to the entire region. This clearly leads to unreasonable values, such as abnormally low emissivity and high albedo, across entire regions for certain countries. The former issue is not only true for CLMU, but also for urban emissivity constraints in regional models like WRF (Chakraborty et al., 2021). For instance, the minimum roof emissivity in J2010 is as low as 0.04 in regions like Mongolia, Kazakhstan, France and Germany (Figure 3, Figure S5) and the roof albedo can be as high as 0.61 for Chili, Argentina, Mongolia and Kazakhstan (Figure 3, Figure S9). These values were derived from specific low-emissivity and high-albedo materials (e.g. zinc/galvanized steel coating), which might be possible for individual buildings, but are highly unlikely for all urban areas in a large region (Chakraborty et al., 2021). Broadcasting to an entire region from sampled material estimates results in an oversimplified representation of urban surfaces. In contrast, the "effective" emissivity retrieved from ASTER GEDv3 (Hulley et al., 2015) in U-Surf is generally higher and more narrowly concentrated, typically between 0.95 and 1.0 across urban facets, with exceptions in a few specific areas. This pattern also aligns with urban canyon characteristics, where the effective emissivity of an urban canyon is slightly higher than the weighted-average values from all individual components due to the "canyon trapping" effects (i.e., increased absorption from reflections between facets) (Harman et al., 2004; Oke et al., 2017). Likewise, we can observe a narrower spread of roof albedo values concentrated between 0.1 and 0.3 across countries, which align with the aggregated values from the commonly-used urban roof materials such as tiles (0.10 - 0.35), shingles (0.05 - 0.25), and slate (0.08 - 0.18) (Oke et al., 2017). Our results confirm that the blue/clear sky albedo (total albedo for shortwave radiation) calculated in U-Surf, an interpolation between white- and black-sky albedo (Liang et al., 1999), represents the real-world conditions more accurately.

The morphological parameters in our dataset also provide more reasonable estimates of both mean values and variability. The four morphological parameters follow similar trends with J2010 in their variations across urban
density types. For example, the roof fractions (pervious fractions) are generally higher (lower) in TBD locations identified in J2010, and decrease (increase) as the built density decreases (i.e., HD, MD and LD). However, U-Surf captures much larger variabilities in these parameters compared to J2010, reflecting a more diverse urban morphology. This is again because of the J2010's approach of applying uniform parameter values based on selected representative buildings in a region. This approach not only fails to represent the granular spatial variability in a region, but also
easily skews the estimates. For example, J2010 reported an unrealistically high roof fraction of 0.8 for the MD class over Brazil; whereas U-Surf presents a more realistic roof fraction predominantly ranging between 0.03 and 0.14, with a median value of 0.07 for this region, which aligns more closely with observations. Note that the median values of the four morphological parameters in U-Surf raw data (black boxes in Figure 3) are generally lower (or higher in "Pervious fraction") than TBD, HD, MD and LD categories (blue boxes). This is because U-Surf raw covers more
pixels than the sum of locations identified as TBD, HD, MD and LD in J2010, most of which are sparsely built landscapes. In fact, the less densely built urban areas dominate the global urban landscapes. The four density classes TBD, HD, MD, and LD in J2010, for example, account for 0.022%, 5.85%, 23.76%, and 70.37% of all urban grids, respectively.

The $H/W$ values in U-Surf are concentrated within ranges of 0.6-1.4, 0.2-0.8, and 0.1-0.4 for TBD, HD, and MD locations identified in J2010, respectively. These values are close to real-world observations which typically vary between 0.5 and 2 at the neighborhood scale (Vardoulakis et al., 2003). Note that high $H/W$ values are very rare at 1-km resolution in real cases. Only very densely built central metropolitan areas (such as the lower Manhattan area in New York City, US) exhibit ratios exceeding 1. These occasions, however, usually only constitute a small proportion.
This explains why the overall raw U-Surf $H/W$ values are mostly concentrated between 0.06 and 0.5. We note that in very rare cases, there are some very high roof fraction numbers ($\geq 0.9$) in U-Surf which nevertheless are not located in central metropolitan areas. These outliers are places with relatively large roof cover but very small "urban" impervious areas (such as, near the edge of a suburban area, or located in small-size dispersed town areas).

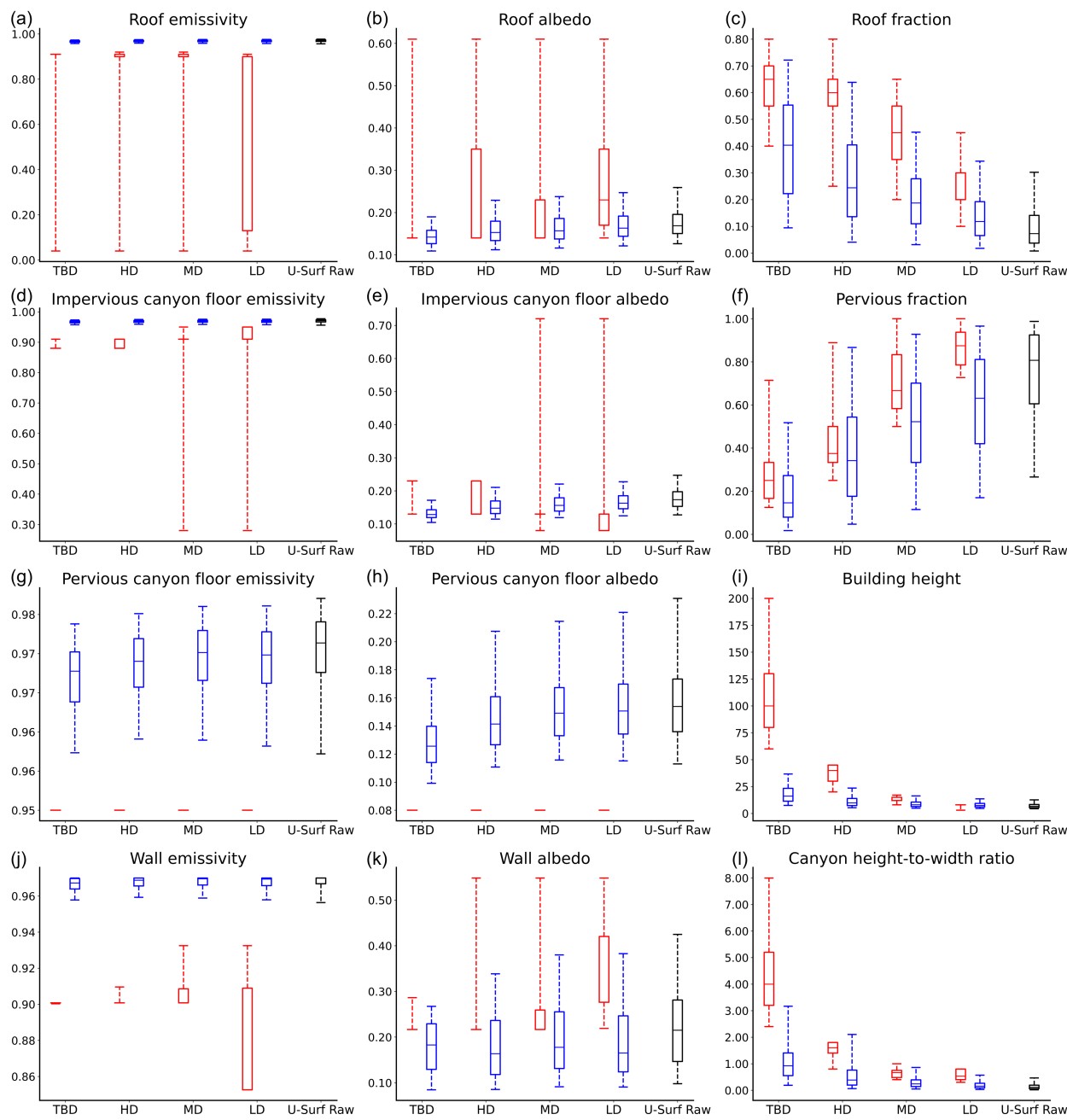

**Figure 3. Distribution of urban surface properties at four density classes locations** (Oleson and Feddema, 2020)**: Tall Building District (TBD), High Density (HD), Medium Density (MD), and Low Density (LD), compared with raw U-Surf data.** Red bars represent CLMU values (discrete, 33 regions), and blue bars show new U-Surf values (continuous, 1 km) extracted from grids identified as TBD, HD, MD and LD per J2010's definition. The black bars show the distribution of 1 km U-Surf raw data. Box and whisker plots show the 25th, median and the 75th (the bottom, middle and top horizontal bars), and extend to the 5th and 95th quantiles.

## 3.2 Enhanced urban surface properties

In this section, we present selected radiative and morphological parameters as illustrative examples to demonstrate the improvement of the new urban surface dataset in spatial heterogeneity, granularity, accuracy, and broader

applicability from global to city scale. The global maps of the complete list of parameters can be found in SI (Figures S5-S26).

U-Surf exhibits significant advancements in capturing the spatial heterogeneity and granularity, which is a crucial improvement over traditional categorical urban classifications, such as density classes used in CLMU and LCZs used in state-of-the-art mesoscale models. The dataset's fine-scale resolution reveals detailed variations in both urban radiative and morphological parameters (Figure 4, Figures S5-S17). For instance, in J2010, the emissivity of pervious canyon floors is uniformly set at 0.95 globally to represent a typical value for vegetation using a bulk parameterization scheme (Oleson et al., 2010), and roof albedo is limited to 11 distinct values (Figures 4a and 4b). These discrete values lead to oversimplifications that fail to represent the critical variations in urban areas, potentially affecting the accuracy of cross-sample variability in urban climate states. Conversely, U-Surf data shows clear variability both within and across regions (Figures 4c and 4d). In general, albedo exhibits greater variability than emissivity across different facets (Figures S5d-S12d). The albedo of impervious canyon floor is comparable with the pervious one, while roof and wall albedo tend to be higher, especially in the city center with densely-built tall buildings (Figures S9d-S12d). In New York City, for example, the mean albedo values are 0.13 for both impervious and pervious canyon floors, while roof albedo averages 0.16 and wall albedo is even higher at 0.22. This pattern is consistent with the fact that commonly used road pavement materials, such as asphalt and concrete, exhibit similarly low albedo values when compared to urban vegetated surfaces like parks and lawns (Oke et al., 2017). Moreover, roofing materials in metropolitan areas often feature higher reflectivity to reduce heat absorption by buildings (Jia et al., 2024), further contributing to the observed differences in albedo. These variations not only reflect differences in materials used but also adaptation strategies to local climate conditions, thereby providing more insights into local climate-sensitive urban design practices.

On the global scale, U-Surf also reveals high-level distinct spatial patterns that correspond to the varying stages of urban development across regions (Figure 5). In the Global North, particularly in Europe and United States, urban areas typically exhibit higher building density (roof fraction × urban percentage), greater average building height, and higher average canyon height-to-width ratio. These characteristics are indicative of more developed urban form and well-established infrastructure, often driven by the need to accommodate growing populations in limited spaces. For instance, metropolitan centers (e.g. Manhattan, New York City, USA; Quartiers 1-4, Paris, France) in these areas frequently exceed 30-40% roof coverage, with average building heights surpassing 30 meters. In contrast, the Global South (e.g., Latin America, Africa, and Central Asia) generally exhibit lower values for these parameters. For example, building density in these regions are 38.59%, 46.46%, 88.71% lower, respectively, than in the United States. Similarly, their median building height is 11.94%, 31.65%, 12.75% lower, respectively, than in Europe. Consequently, their median canyon height-to-width ratios are 29.88%, 37.18%, 23.99% lower, respectively, than those in Europe. However, this trend is rapidly changing in emerging economies, including India and Brazil, where cities are experiencing swift urban growth. For instance, rapidly urbanizing places such as Delhi, India and Sao Paulo, Brazil have demonstrated tall and densely built environments, where Delhi has a roof fraction of 31.02% and building heights

of 12.63m, while Sao Paulo has a roof fraction of 49.42% and building heights of 13.87m, all of which exceed the 75th percentile in the global distribution (Figure 3c). Additionally, regions such as East Asia exhibit urbanization

patterns that are more akin to those in North America and Europe, characterized by high roof fractions (e.g., Figure S4a) and significant vertical development. For example, many cities in Eastern China have exhibited city-wide average roof fractions above 14% and building heights exceeding 13m, reflecting rapid industrialization and economic growth that have rapidly transformed the urban landscape over the past few decades (Cai et al., 2022). These observations further demonstrate the fidelity of U-Surf to reveal globally comparable yet regionally nuanced urbanization

representations, which are essential for understanding geographical disparities and advancing region-specific sustainable urban development.

At a more localized level, U-Surf uncovers intriguing patterns within countries and even individual cities, offering insights into the complex interactions between urban morphology and local climate conditions. For instance, in the

Southwestern United States (e.g., California, Arizona), Northern Africa countries like Egypt and Tunisia, along with Northeastern China, U-Surf captures lower pervious canyon floor emissivity (below 0.93) and higher roof albedo (above 0.25) (Figures 4c and 4d), reflecting the potential impact of arid conditions and the use of high-albedo materials for heat adaptation in hot climates. Furthermore, U-Surf highlights regional differences in surface morphological properties (Figure 5), which play crucial roles in determining local urban climates. Building heights are notably higher

along the coasts and in southern regions of Contiguous United States (CONUS), with cities like New York, Chicago, and Miami showing exceptionally high values (> 100m) and corresponding high canyon height-to-width ratios (> 2) in city cores. These cities also exhibit high roof fractions, showing more clustered building patterns in city centers, with density decreasing outwards (Figure S4b). In densely populated developing countries like India and China, high roof fractions exceeding 40% are observed, particularly in regions such as the Indo-Gangetic Plain and the area

spanning from the Bohai Economic Rim to the Yangtze River Basin (Figure S4a). In underdeveloped regions of South America and Africa (e.g., Bolivia, Chad) with widely dispersed urban areas, buildings are more sparsely distributed, typically concentrated within fewer metropolitan areas. It is interesting to note that the high-resolution U-Surf data even captures the very densely populated informal settlements (such as the slum areas in Delhi) where buildings are tightly packed and often overcrowded (characterized by high roof fraction and population density) (Figures S4c and

S4d). This illustrates the potential use of U-Surf as a valuable tool to better inform socioeconomic disparities in environmental and climate hazards within cities, currently difficult to do using process-based models (Chakraborty et al., 2023; Zhao et al., 2021).

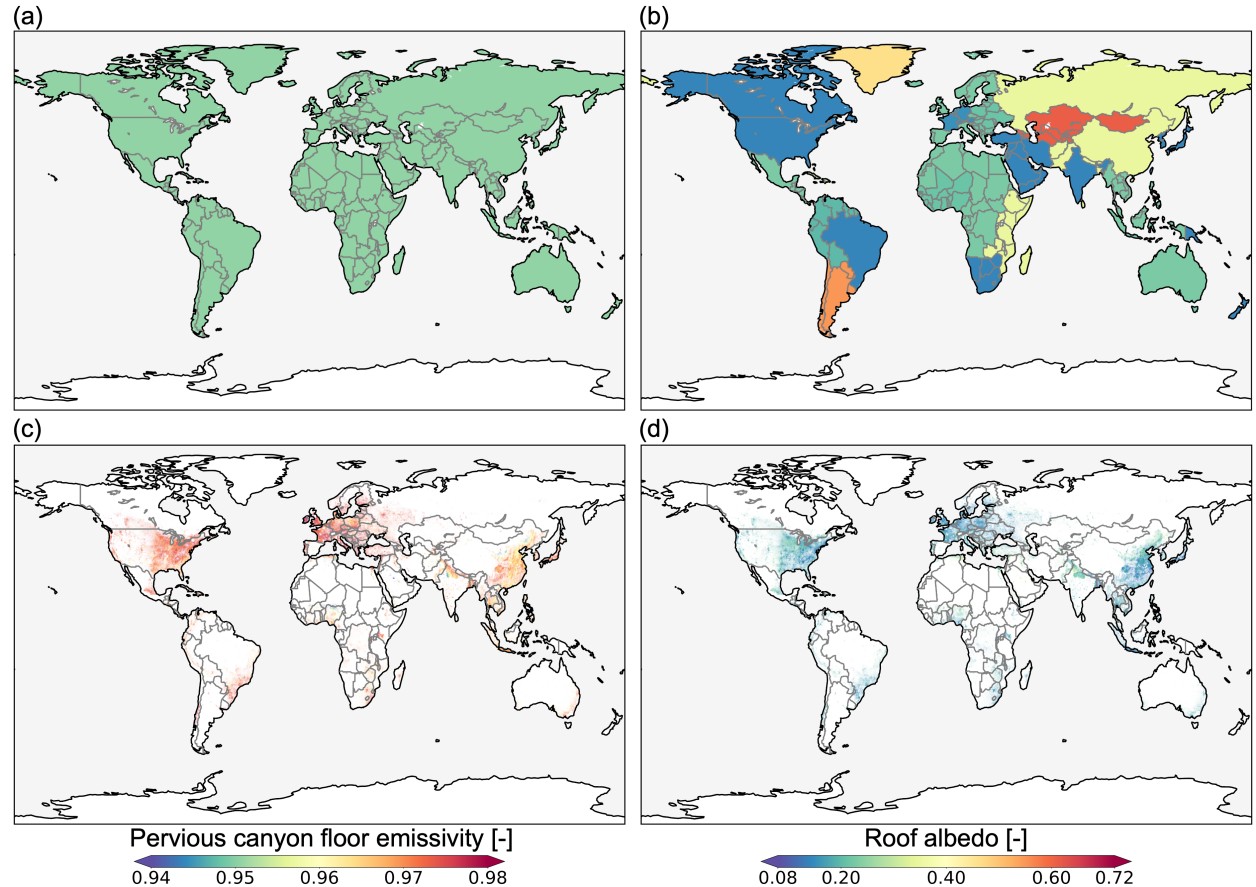

**Figure 4. Global-scale comparison between the default CLMU and U-Surf parameters.** (a, b) Discrete pervious canyon floor emissivity [unitless] and roof albedo [unitless] over 33 regions (area-weighted averages across TBD, HD and MD) in CLMU; (c, d) 1-km continuous pervious canyon floor emissivity and roof albedo in U-Surf. Each column shares the same color scale range but note that default CLMU parameters only have discrete values over 33 regions and 3 density classes.

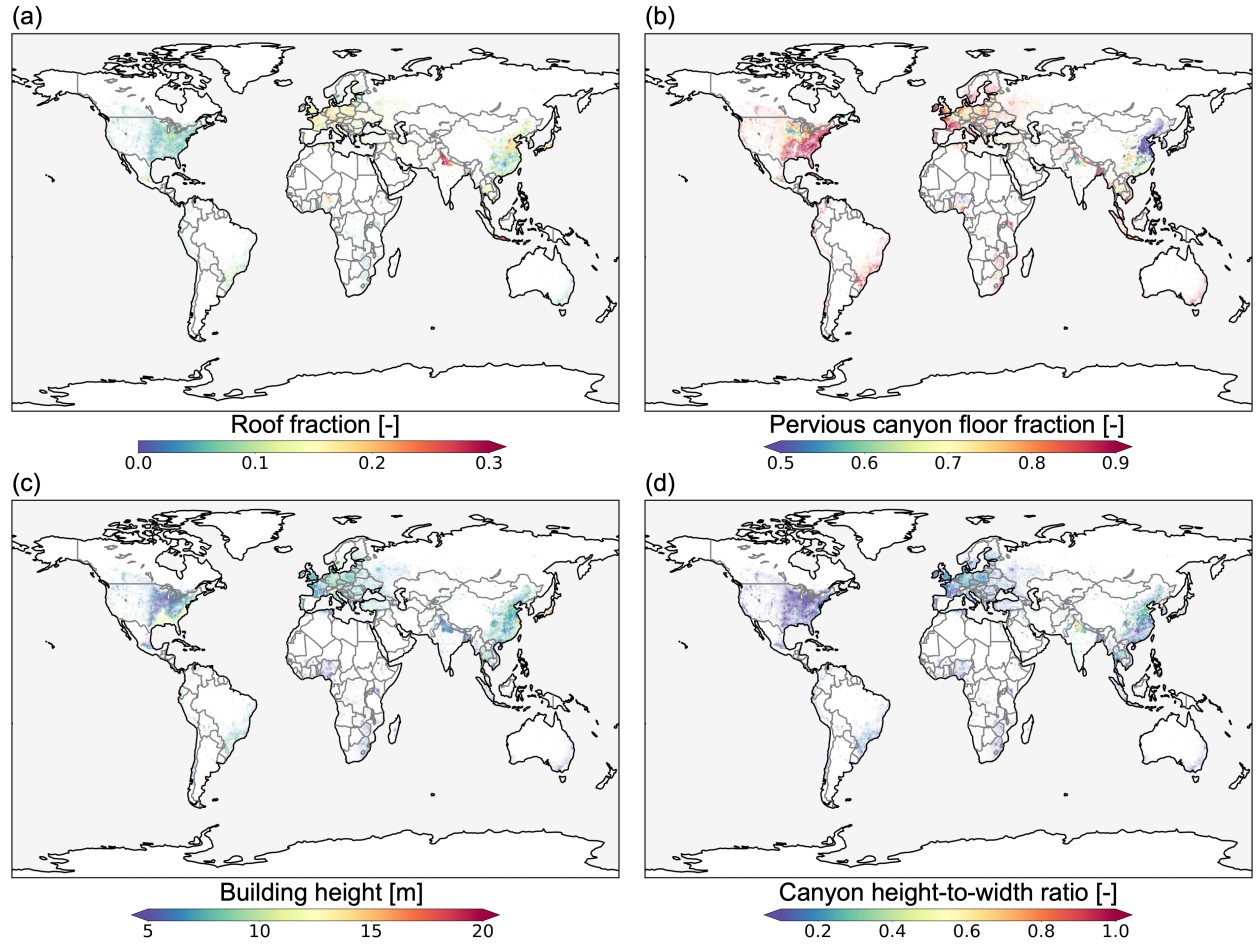

**Figure 5. Global spatial distribution of U-Surf morphological parameters.** (a) roof fraction [unitless], (b) pervious fraction [unitless], (c) building height [m], (d) canyon height-to-width ratio [unitless].

### 3.3 Improved urban representation across scales

The high-resolution U-Surf data enables intra- and inter-city comparisons in global-scale urban climate modeling in an unprecedented way. To illustrate this point, we identified two cities with similar background climates: Chicago, IL, USA, and Seoul, South Korea (Figure 6), both of which are classified as *Dfa* under Koppen-Geiger climate classification (humid continental climate with hot summers) (Beck et al., 2018). Because of the coarse-resolution urban surface input in J2010, these two cities share the exact same roof-specific parameters of the MD class. However, U-Surf reveals distinct contrasts in their radiative and morphological properties. Chicago, which has a history of applying heat mitigation strategies such as cool roofs (Mackey et al., 2012; Zhao et al., 2014), demonstrates higher roof emissivity and albedo, with average values of 0.972 and 0.175, respectively, compared to Seoul's 0.955 and 0.114. As to intra-city variations, Chicago's urban form is characterized by a higher concentration of buildings, with an average roof fraction of 0.284, in the northern part of the city. High-rise buildings or skyscrapers are predominantly clustered around Lake Michigan and the Chicago Loop area. On the contrary, Seoul exhibits a more dispersed urban structure, with buildings spread more widely across the city. Such detailed representation facilitates comprehensive attribution and sensitivity analyses, permitting the examination of how individual parameters, such as emissivity,

albedo and building height, can alter the city microclimate and further influence the role of cities in local to global climate change scenarios (Krayenhoff et al., 2018, 2021; Zhao et al., 2017) and potentially informs more actionable climate adaptation and mitigation strategies.

U-Surf demonstrates a remarkable ability to capture the spatial heterogeneity and textural details of global urban landscapes across scales. To demonstrate this point, we aggregated the 1km U-Surf data to coarser resolutions of 0.125° and nominal 1° (a typical resolution that ESMs are run at) to compare with J2010 side by side. More detailed information about the aggregation process can be found in Supplementary Text S1, Table S1 and Figure S27. For

illustrative purposes, only the comparisons of $H/W$ are shown here. Our results demonstrate that U-Surf represents the detailed urban form considerably well, even at much coarser resolutions. The spatial variability and urban texture are well preserved at global (Figure 7d), national (Figure 7e), and city (Figure 7f) scales. This further demonstrates the adaptability and application of U-Surf in multi-scale, cross-scale urban modeling studies, with potential usage in regionally refined models (RRM) such as E3SM-RRM (Tang et al., 2023a) as well as the variable-resolution models

(Huang et al., 2016) like Multi-Scale Infrastructure for Chemistry and Aerosols (MUSICA; Pfister et al., 2020), where seamless transitions between different spatial scales are crucial for comprehensive and coherent analysis.

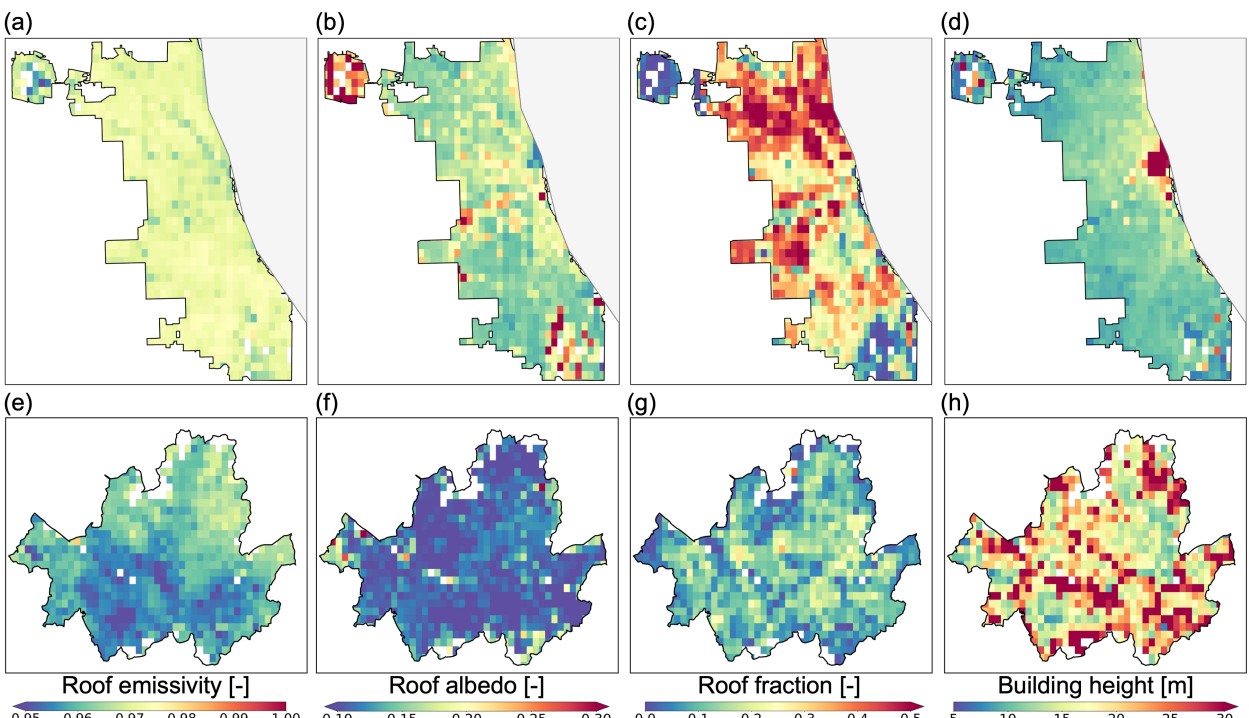

**Figure 6. Spatial distributions of roof emissivity [-], roof albedo [-], roof fraction [-], building height [m] in (a-d) Chicago, USA and (e-h) Seoul, South Korea.** Each pair of panels within the same column shares a consistent color scale.

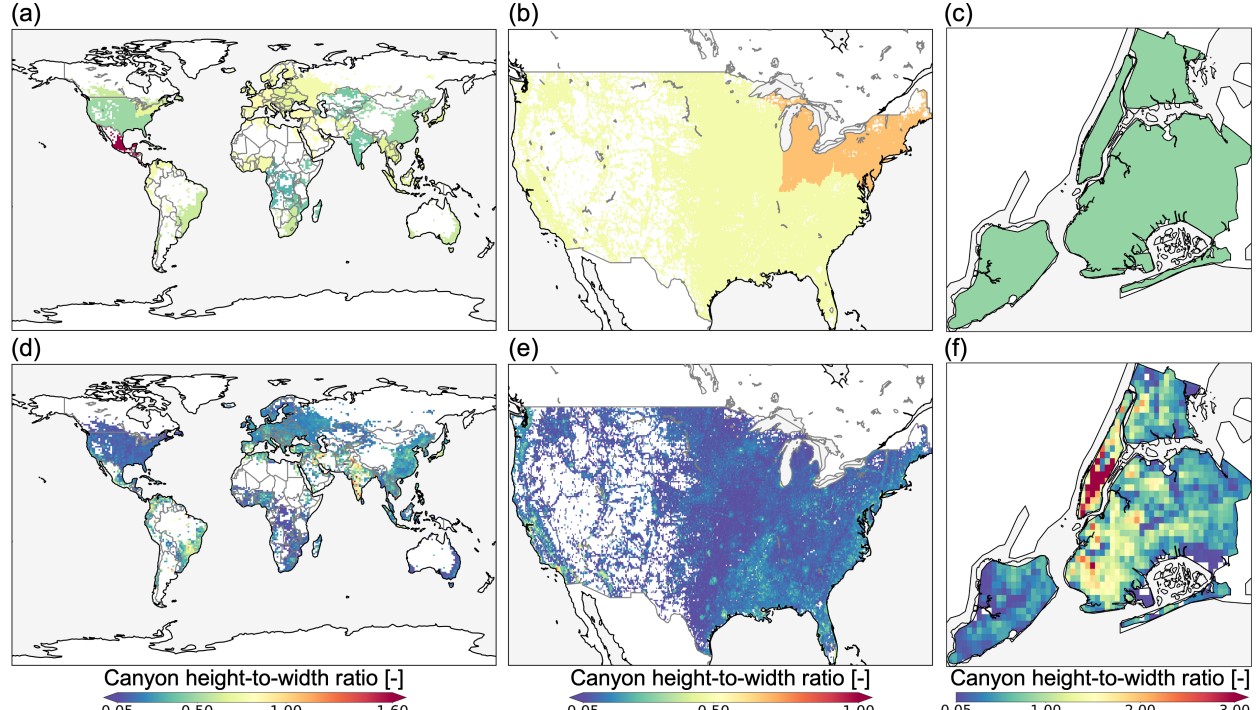

**Figure 7. Spatial variability of (a-c) area-weighted averages from CLMU and (d-f) U-Surf surface dataset.** The parameter shown here is canyon height-to-width ratio. The spatial resolution from left to right: global nominal 1-deg, 0.125-deg over CONUS, and 1km in New York City, US. Panels in the same column share a colorbar at the bottom of the column. Different colorbar ranges are used to help visualize the distributions across scales.

### 3.4 Accuracy assessment and uncertainty

For the derived morphological parameters, we conducted a thematic validation based on two recently available, observation-based datasets, Urban-PLUMBER and WSF-3D. WSF-3D is a high-resolution (~ 90 m at the equator) global dataset that provides detailed three-dimensional information on building fraction, height, and volume, derived from satellite imageries, offering crucial insights into urban structures and their spatial distribution across the globe (Esch et al., 2022). We compared the roof fraction and height at 1 km resolution across WSF-3D's 17 validation sites. The Urban-PLUMBER project primarily aims to enhance the understanding of the accuracy of current urban climate models and has also produced a harmonized dataset of quality-controlled and gap-filled observations from 21 urban flux tower sites across different climate zones and urban built environments (Lipson et al., 2022). We compared all four morphological parameters across these sites by using neighboring pixels around the flux towers to evaluate against the site-specific information.

The roof fraction showed strong agreement across the reference sites in both WSF-3D and Urban-PLUMBER, with low mean absolute errors (MAEs) of 0.076 and 0.081 (Figure 8a). Similarly, the pervious fraction also aligned well at most Urban-PLUMBER sites, with a mean MAE of 0.124 (Figure 9a). Some discrepancies were observed in building height (MAE=5.918m and 7.446m, Figures 8c and 8d) and canyon height-to-width ratio (MAE=0.387, Figure 9c). These discrepancies are primarily attributed to the disparity between the neighborhood-scale values captured by flux

towers, typically representing areas within several hundreds of meters, and the 1 km-resolution averaged values.

Detailed values at each individual site can be found in Tables S3 and S4.

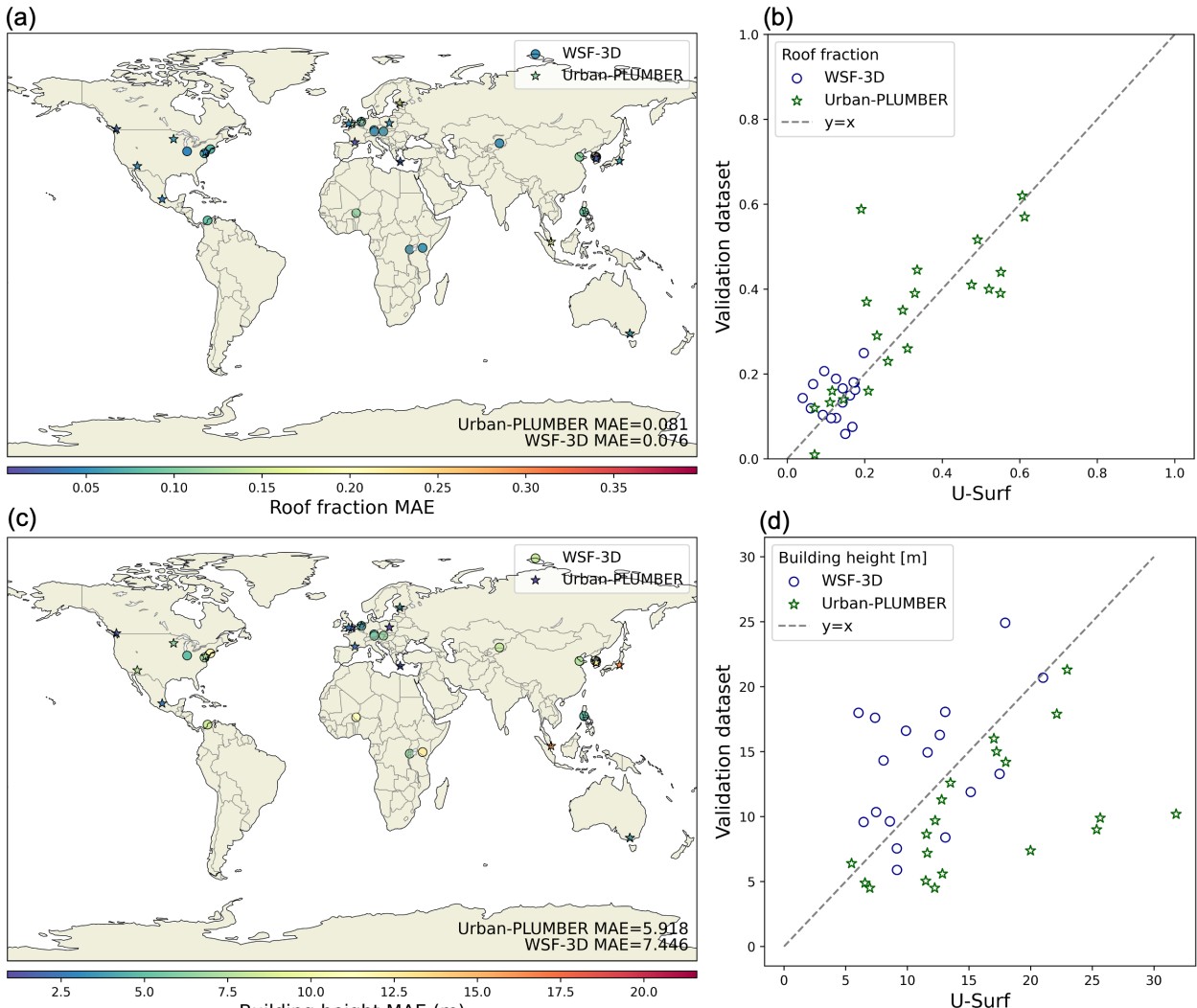

**Figure 8. Comparison of two morphological parameters: (a-b) roof fraction, and (c-d) building height, evaluated against 21 Urban-PLUMBER and 17 WSF-3D sites.** The numbers labeled on the bottom right corner of (a) and (c) indicate the average
mean absolute errors (MAEs) across sites. The blue points in (b) and (d) represent the city-scale average values at WSF-3D sites. The green stars in (b) and (d) are the site-specific values from Urban-PLUMBER metadata.

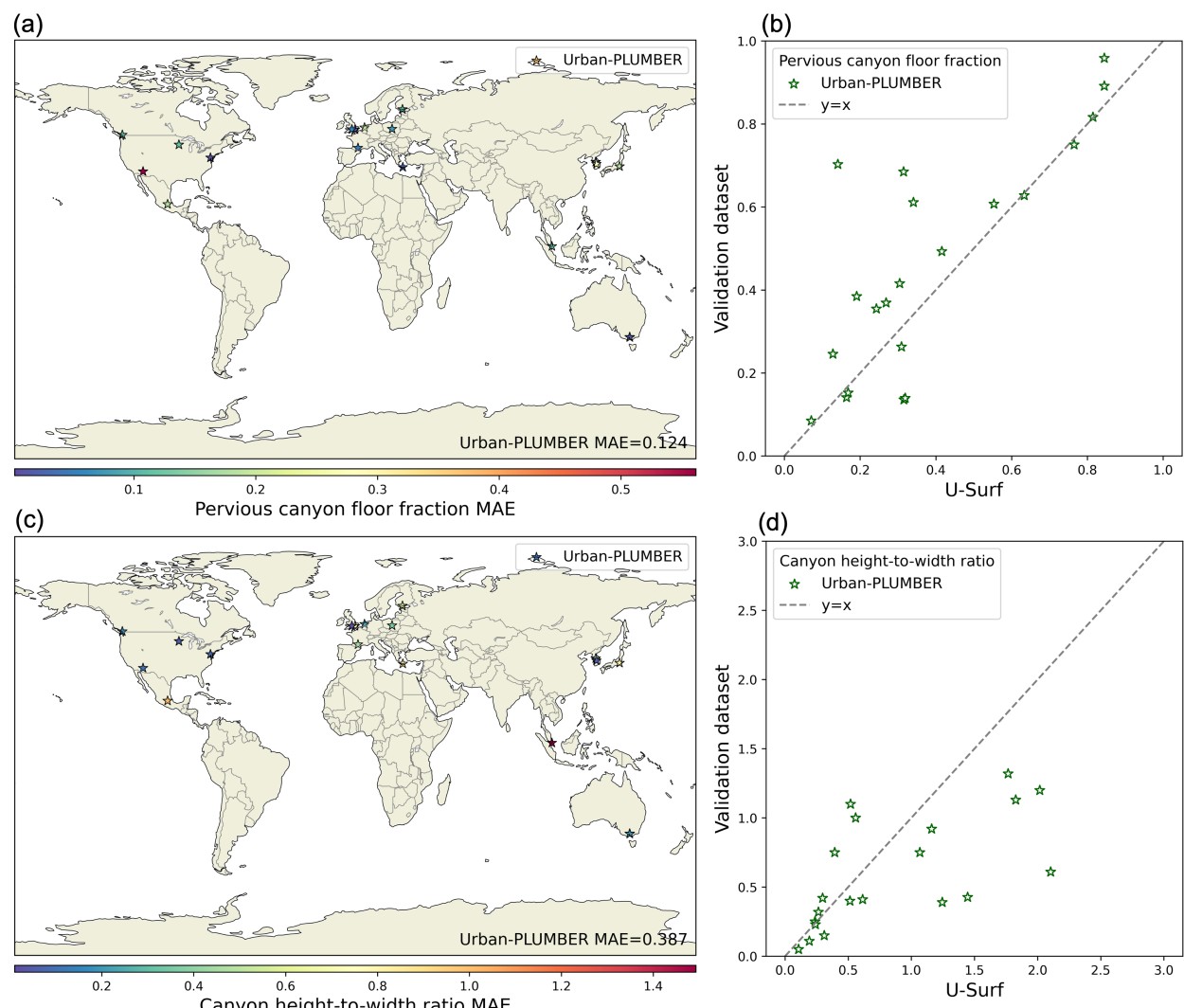

**Figure 9. Comparison of two morphological parameters: (a-b) pervious canyon floor fraction, (c-d) canyon height-to-width ratio, against 21 Urban-PLUMBER sites.** The numbers labeled on the bottom right corner of (a) and (c) represent the average mean absolute error across sites.

As discussed briefly in Sect. 2.4, U-Surf's parameters are inherently influenced by the uncertainties embedded in the synthesized data sources and uncertainty propagation during calculations. To systematically evaluate the uncertainties in final U-Surf parameters, we first documented the available validation approaches, as conducted by the development teams, and associated uncertainties for all input data sources in Table 3. Based on these numbers, we then employed the Monte Carlo simulation approach to quantify the final uncertainties in all our derived urban surface parameters in U-Surf (see Supplementary Text S2).

**Table 3. Validation and uncertainty analysis of synthesized data products.**

| Dataset | Source | Validation | Uncertainty |
|---|---|---|---|
| **ESA Landcover 2021 v200** | Zanaga et al., 2022 | Validated using Copernicus Global Land Service-Land Cover Validation dataset | Global accuracy of 76.7% $\pm$ 0.5%<br>User's accuracy: Treecover 80.0% $\pm$ 0.7%, bare/sparse vegetation: 92.1% $\pm$ 0.9%, shrubland: 49.1% $\pm$ 2.1%, grassland:71.9% $\pm$ 1.0%, built-up: 65.9% $\pm$ 3.3% |
| **Microsoft Global Building Footprints** | Microsoft, 2022 | Evaluated on a set of building polygon labels for each region based on Bing Maps including Maxar and Airbus between 2014 and 2021 | Precision: 92.2% (Caribbean) -- 97.17% (Central Asia)<br>False positive rate: Mexico 0.1%, North America 1.0%, Africa 1.1%, Australia 1.1%, Europe 1.4%, South Asia 1.4%, South America 1.7%, Caribbean 1.8%, Middle East 1.8%, Central Asia 2.2%, Indonesia 2.98% |
| **East Asia Building Footprints** | Shi et al., 2024 | Validated in sampled Chinese cities with manual annotation, compared against OSM building data and regional roof vectors | Accuracy: 89.63% (average)<br>F1 score: 82.55% |
| **ASTER Global Emissivity Dataset v3** | Hulley et al., 2015 | Validated against lab measurements & MODIS C4, C5 emissivity (2000-2008) over selected 4 sites | RMSE of 0.41%, 0.84%, 0.87%, 0.95% at four sites: Algodone Dunes, Namib, Senegal Basin and Rub Al Khali |
| **Broadband ASTER Emissivity** | Malakar et al., 2018; Ogawa et al., 2008 | Validated against 305 samples from ASTER spectral library covering the wavelength ranging from 2-15$\mu m$ | $R^2 = 0.913$, $RMSE = 0.011$ |
| **Sentinel-2 Albedo** | Lin et al., 2022 | Validated against ground measurements & MODIS satellite product at local flux sites | Overall across 5 land cover types: $R^2 = 0.94$, $RMSE = 0.030$<br>Deciduous broadleaf forest: $R^2 = 0.58$, $RMSE = 0.027$<br>Evergreen needleleaf forest: $R^2 = 0.72$, $RMSE = 0.028$<br>Grassland: $R^2 = 0.95$, $RMSE = 0.032$<br>Open shrubland: $R^2 = 0.92$, $RMSE = 0.026$<br>Urban: black sky albedo $R^2 = 0.90$, $RMSE = 0.0185$, white sky albedo $R^2 = 0.87$, $RMSE = 0.0205$ (average), blue sky albedo $RMSE = 0.0154$ |
| **Sentinel-2 Narrow-to-broadband Albedo** | Bonafoni & Sekertekin, 2020 | Validated against ground measurements at selected sites | $R^2 = 0.77$, $RMSE = 0.023$ when compared against six Surface Radiation Budget Network stations measurements during 2018-2019<br>$R^2 = 0.98$, $RMSE = 0.021$ when compared against albedometer measurements at 18 Perugia sites, summer 2016 |
| **Building Height** | Che et al., 2024 | Validated against various reference dataset and selected cities from Google Earth Pro | Varying across 33 subregions<br>$R^2$: 0.66 (Europe) -- 0.96 (South America)<br>RMSE: 1.92m (South America) -- 14.60m (Japan, North and South Korea) |
| | Li et al., 2022 | Evaluated on the validation set, compared against WSF-3D | Global: $R^2 = 0.73$, $RMSE = 2.56$<br>Canada and USA $R^2 = 0.72$, $RMSE = 2.01$, China $R^2 = 0.49$, $RMSE = 4.94$, Europe $R^2 = 0.68$, $RMSE = 2.35$, South Asia $R^2 = 0.47$, $RMSE = 1.79$, Latin America $R^2 = 0.60$, $RMSE = 2.86$, Middle-East and Northern Africa $R^2 = 0.75$, $RMSE = 2.92$, Oceania $R^2 = 0.70$, $RMSE = 1.58$, Russia and Central Asia $R^2 = 0.48$, $RMSE = 2.78$, Southeast Asia $R^2 = 0.62$, $RMSE = 1.50$, Sub-Saharan Africa $R^2 = 0.63$, $RMSE = 1.35$ |

| AC Penetration Rate | Li et al., 2024b | 35 countries/regions were directly collected; additional linear model was built to map other 34 regions/countries and sub-country level data | Linear model $R^2 = 0.9$, $RMSE = 11.5\%$, $MAE = 8.5\%$ |
| --- | --- | --- | --- |

Specifically, three datasets used to differentiate roofs, impervious and pervious canyon floors demonstrate high global classification accuracy. The 10m-resolution ESA land cover (Zanaga et al., 2022) was validated using updated Copernicus Global Land Service-Land Cover Validation (CGLS-100) dataset. The global overall accuracy across all land cover types is 76.7±0.5%. The confidence intervals for specific land cover types are 3.3% for built-up surface and average 1.2% for pervious canyon (the average value of tree cover, grassland, shrubland, bare soil). The MS-BFP

data (Microsoft, 2022) were evaluated using building polygon labels from Bing Maps, including Maxar and Airbus data. The precision of semantic segmentation (i.e., building pixel detection) showed regional variations with the lowest false positive rate of 0.1% in Mexico and highest false positive rate of 2.98% in Indonesia. The EA-BFP (Shi et al., 2024) were validated in sampled Chinese cities with manual annotation, compared against OSM building data and regional roof vectors (Zhang et al., 2022). It has an overall average accuracy of 89.63% and F1 score of 82.55%. The

primary data source of building height (Chet et al., 2024) underwent rigorous validation against various reference height datasets and selected cities from Google Earth Pro. The validated results showed $R^2$ values ranging from 0.66 (Europe) to 0.96 (South America) and Root Mean Squared Error (RMSE) from 1.9m (South America) to 14.6m (Japan, North and South Korea) across different subregions. The supplementary dataset (Li et al., 2022) was also validated and compared against WSF-3D, yielding a global RMSE of 2.56m, with the lowest RMSE of 1.35m in Sub-Saharan

Africa and the highest RMSE of 4.94m in China.

All remote sensing products and algorithms used to derive radiative properties were validated against ground measurements with high credibility. ASTER GEDv3 (Hulley et al., 2015) was compared with MODIS Collection 4 & 5 Emissivity and validated against lab measurements at four large sand dune fields, yielding a relatively low average

RMSE of 0.077. The broadband emissivity regression algorithm (Eq. 1) was validated against ASTER spectral library covering the wavelength ranging from 2-15μm, yielding the $R^2$ of 0.913 and RMSE of 0.011 (Malakar et al., 2018; Ogawa et al., 2008). The 10m land blue-sky albedo (Lin et al., 2022), retrieved from Sentinel-2 surface reflectance, was validated against local flux tower measurements, achieving an overall $R^2$ of 0.94 and RMSE of 0.03 across five land cover types. The RMSE ranges from around 0.0154 for urban areas (see Supplementary Text S2 for detailed

calculation) to 0.032 for grassland. In addition, the narrow-to-broadband algorithm (Bonafoni and Sekertekin, 2020) demonstrated a $R^2$ of 0.77 and RMSE of 0.023 when compared against the ground measurements at six Surface Radiation Budget Network (SURFRAD) stations. It also showed a $R^2$ of 0.98 and RMSE of 0.021 when compared against albedometer measurements at eighteen Perugia sites (Bonafoni and Sekertekin, 2020).

The primary source of uncertainty in the AC adoption rate (Li et al., 2024b) stems from the linear model that correlates AC adoption rate with the number of AC units per household. The linear model with saturation effect has an $R^2$ of 0.9 (p < 0.001), RMSE of 11.5 and MAE of 8.5 (both in the unit of %).

Using these documented uncertainties, we conducted Monte Carlo simulations with 1,000 trails of randomly perturbed
input parameters based on 10,000 randomly selected samples across 10 countries (Table S2) to quantify the uncertainty
of error propagation through our data synthesis and processing (Supplementary Text S2). The resulting 95%
confidence intervals for all parameters across all sampled regions and global averages are presented in Table 4. These
intervals provide the expected error/uncertainty ranges for our final estimates. Overall, the uncertainties propagated
through our data synthesis and processing align closely with those in the input data and remain relatively small – partly
due to spatial upscale from finer resolutions to 1 km – which confirms the robustness of our methodology.

**Table 4. Estimated 95% confidence intervals (±) by Monte Carlo simulations across all regions.**

| Continent | | | North America | | South America | | Europe | | Asia | | Oceania | Africa | |
|---|---|---|---|---|---|---|---|---|---|---|---|---|---|
| Country | | | United States | Mexico | Argentina | Bolivia | France | Poland | China | Malaysia | Australia | Nigeria | Average |
| Radiative | Emissivity | Roof | 0.0443 | 0.0316 | 0.0342 | 0.0477 | 0.0386 | 0.0381 | 0.0428 | 0.0359 | 0.0542 | 0.0432 | 0.0411 |
| | | Impervious | 0.0369 | 0.0326 | 0.0289 | 0.0443 | 0.0479 | 0.0450 | 0.0394 | 0.0432 | 0.0616 | 0.0560 | 0.0436 |
| | | Pervious | 0.0181 | 0.0137 | 0.0169 | 0.0133 | 0.0149 | 0.0159 | 0.0164 | 0.0135 | 0.0126 | 0.0202 | 0.0156 |
| | | Wall* | 0.0443 | 0.0316 | 0.0342 | 0.0477 | 0.0386 | 0.0381 | 0.0428 | 0.0359 | 0.0542 | 0.0432 | 0.0411 |
| | Albedo | Roof | 0.0086 | 0.0043 | 0.0034 | 0.0096 | 0.0047 | 0.0045 | 0.0169 | 0.0041 | 0.0080 | 0.0069 | 0.0071 |
| | | Impervious | 0.0067 | 0.0058 | 0.0044 | 0.0099 | 0.0101 | 0.0082 | 0.0056 | 0.0082 | 0.0129 | 0.0118 | 0.0084 |
| | | Pervious | 0.0020 | 0.0009 | 0.0012 | 0.0010 | 0.0010 | 0.0011 | 0.0012 | 0.0007 | 0.0007 | 0.0019 | 0.0012 |
| | | Wall* | 0.0086 | 0.0043 | 0.0034 | 0.0096 | 0.0047 | 0.0045 | 0.0169 | 0.0041 | 0.0080 | 0.0069 | 0.0071 |
| Morphological | | Urban percentage | 0.0029 | 0.0039 | 0.0048 | 0.0033 | 0.0031 | 0.0031 | 0.0047 | 0.0034 | 0.0023 | 0.0023 | 0.0034 |
| | | Roof fraction | 0.0019 | 0.0029 | 0.0024 | 0.0029 | 0.0045 | 0.0046 | 0.0061 | 0.0024 | 0.0026 | 0.0087 | 0.0039 |
| | | Pervious fraction | 0.0056 | 0.0060 | 0.0059 | 0.0076 | 0.0073 | 0.0085 | 0.0104 | 0.0049 | 0.0074 | 0.0107 | 0.0074 |
| | | Canyon height to width ratio | 0.0751 | 0.1702 | 0.2130 | 0.1307 | 0.1592 | 0.1585 | 0.2156 | 0.1058 | 0.0681 | 0.1052 | 0.1401 |
| | | Building height (m) | 3.8092 | 5.0254 | 5.0589 | 5.2250 | 4.4660 | 4.5200 | 8.2679 | 2.8700 | 3.0307 | 2.5947 | 4.4868 |

*Wall radiative parameters were processed by directly utilizing the roof parameters, resulting in the same uncertainty range.

**4 Broader implications of U-Surf**

The U-Surf dataset significantly advances the development of ultra-high resolution urban-resolving process-based
ESMs and RCMs. Its high-resolution capabilities allow for a detailed and refined representation of urban areas,
breaking away from the limitations of previous models that relied on coarse regional divisions and outdated
classifications. By integrating the latest global data sources, U-Surf provides global continuity and local granularity

in urban surface representation. This enhanced representation shows promise in correcting systematic biases in current models and improving their modeling accuracy and predictability. For example, a recent study finds that the simulated urban heat island (UHI) effects tend to be overestimated in CESM2 (Liu et al., 2024). To test the effects of U-Surf, we have run two preliminary land-only CESM2 simulations (0.9375°×1.25°) spanning from 2010 to 2014 with the default urban surface data and U-Surf, both forced by bias-corrected ERA5 (Cucchi et al., 2020). We find that this

overestimation is largely reduced by an average of 0.176K in annual canopy UHI (CUHI) over China, due to the widespread cooling trend in urban near-surface air temperatures (Figure S28). This improvement aligns with Liu et al. (2024)'s findings that CESM2 overestimates CUHI in China by +0.127°C. Moreover, the remote-sensing-based methodology offers a unique capability to track the quantitative evolution of urban canopy parameters (UCPs) over time, a level of detail that is difficult to extract from traditional classification methods.


While developed with the architecture of Earth system models in mind (namely CLMU and its versions used in various ESMs), U-Surf can be adapted to other UCMs, such as those embedded in RCMs like WRF, and atmospheric chemistry models such as MUSICA (Pfister et al., 2020; Tang et al., 2023b). Its scalability enables its use in studies ranging from local-scale high-resolution applications to regional and global-scale analyses. Incorporating detailed

fine-resolution UCPs (e.g., plan area fraction $\lambda_p$, frontal area index $\lambda_F$), as demonstrated in WRF studies, is essential for accurately modeling urban climate dynamics (Best and Grimmond, 2014; Georgescu, 2015; Sharma et al., 2017). U-Surf's application in the next-generation kilometer-scale models could help resolve fine-resolution processes such as convection and advection, further advancing the high-fidelity climate and air quality simulations.

Finally, the implications of U-Surf extend beyond the realm of climate or Earth system modeling. This comprehensive dataset provides essential urban informatics and properties on the global scale that can be directly used as key input features for machine learning models (Chajaei and Bagheri, 2024; Furuya et al., 2023; Li et al., 2023b), making U-Surf a valuable resource for both process-based and data-driven modeling. U-Surf can potentially serve as a powerful tool for researchers, policymakers, and urban planners across multiple disciplines. In studies examining interactions

between urbanization and socio-economic characteristics, for instance, the dataset can be utilized to explore correlations between urban morphology and economic indicators, potentially revealing relationships between building density, green space distribution, and neighborhood income levels (Chakraborty et al., 2022; Wang et al., 2024). In the public health sector, U-Surf could be used to investigate links between urban structural design and air quality (Zhang et al., 2023b; Zhang and Gu, 2013). Moreover, the dataset could be beneficial for emergency management and

disaster preparedness, enabling more accurate risk assessments in densely built areas (Li et al., 2020b; Ma and Mostafavi, 2024).

**5 Limitations and future work**

We note several limitations, which also present opportunities for future improvements. The accuracy of U-Surf is inherently linked to the uncertainties of the synthesized data sources. For instance, the use of Microsoft global building

footprints (mostly 2014-2021, with additional updates up to 2023) may result in missing roofs or land cover

misclassification within certain pixels. One primary challenge arises from integrating datasets with varying spatiotemporal coverage. Most of the datasets we utilized reflect urban surface properties from 2014 to 2021. Although the temporal discrepancy among different data sources may introduce additional uncertainties, given the small changes in built surfaces within this short time span, these uncertainties are likely small. Additionally, the spatial resolution of ASTER GEDv3 data is 100 meters, which could be too coarse to accurately distinguish small individual facets, potentially resulting in mixed facet representation. There is also room for improvements in the remote sensing algorithms used to derive some of the raw surface properties incorporated into U-Surf since they are not always calibrated for urban areas (Chakraborty et al., 2021; Chen et al., 2016), though this is beyond the scope of this study. Lastly, we note that certain U-Surf morphological parameters are constructed on the basis of the 2D infinite-street urban canyon conceptual model used in CLMU. Direct application of those parameters should follow the same conceptual assumptions of the urban geometry. Caution should be given if they are used in more complex representations of the real-world urban landscapes.

We plan to continue improving the U-Surf in future versions through multiple aspects. For example, we anticipate that ongoing efforts and continuing endeavor of urban scientific and remote sensing communities will lead to the emergence of more datasets with higher spatial resolutions and accuracies (e.g., more comprehensive building footprints) to be incorporated or updated in U-Surf. We will also adjust the parameter list to reflect advancements in urban parameterization within RCMs and ESMs. For instance, while urban vegetation is not explicitly represented in the current version of CLMU and most operational RCMs, we can follow similar data pipelines and set of constraints (same land cover data, building footprint estimates, etc.) to develop internally consistent global urban vegetation estimates, known to strongly modulate global inter-sample variability in urban climate signals (Chakraborty and Lee, 2019), for the next-generation UCMs in the future. Lastly, depending upon the availability of data sources and new downscaling approaches, we plan to provide temporally varying urban surface properties, which are important for capturing changes in various urban climate signals over time (Chakraborty and Qian, 2024; Fang et al., 2023; Wu et al., 2024).

**6 Data availability**

The global 1km continuous urban surface property dataset (U-Surf) is publicly available at https://doi.org/10.5281/zenodo.11247598 (Cheng et al., 2024). In addition to the raw dataset at 1km resolution, we also provided the CESM2/E3SM-compatible version at standard resolution (0.9375° × 1.25°) as the ready-to-use input surface dataset for CESM2 simulations. The U-Surf dataset will be incorporated as part of a future release in the high-resolution branch of CTSM (https://github.com/ESCOMP/CTSM).

To facilitate interactive data visualization, query, download, and location-specific analysis, we have further developed a web application using Google Earth Engine (GEE). This interactive platform allows users to explore various urban areas by zooming in on the map, toggle between different parameter layers for comprehensive analysis and extract precise values for all parameters by simply clicking on points of interest or selecting area of interests. The GEE web

application is publicly available at https://ycheng1891.users.earthengine.app/view/global-1km-urban-surface-property-dataset.

The code and intermediate data layers are available from authors upon request.

**7 Conclusion**

Despite recent advances in urban climate model development across scales, one long-standing critical barrier remains: the absence of a complete, fine-resolution, globally consistent, and spatially explicit urban surface property dataset. Existing products relying on broad categorization underscores the challenge of developing an urban representation

that can balance global consistency and local precision. This has been preventing the development of urban-resolving Earth system models for decades, as well as the ultra-high-resolution urban modeling across scales. To address this challenge, we develop a first-of-its-kind global 1km continuous urban surface property dataset – U-Surf. Leveraging recent advancements in remote sensing technologies and machine learning algorithms, U-Surf provides a comprehensive, present-day dataset of urban surface properties that can be used in state-of-the-art ESMs and RCMs.


The high-resolution U-Surf data significantly enhances the urban representation in terms of both spatial heterogeneity and accuracy on the global scale, enables detailed city-to-city comparisons in Earth system modeling, and facilitates high-resolution urban climate modeling across scales. By breaking the constraints of predefined urban density classes, the new dataset provides a more nuanced and accurate representation of urban environments worldwide. The remote-

sensing-based approach captures the actual surface properties as observed from space, accounting for the complex mixture of materials and structures in urban areas that are difficult to illustrate through traditional bottom-up material-based approaches, which provides more effective and accurate urban canopy parameterization compared to the generalization of material-based values used in previous dataset.

The dataset represents a key step forward in advancing the development of ultra-high resolution Earth system modeling. While developed consistent with common ESM architecture in mind, U-Surf can be adapted to quite easily for other models such as weather and regional climate models, and air pollution models, and may be useful inputs for machine learning algorithms. U-Surf also provides useful urban informatics for research and applications across multiple disciplines such as socioeconomics, public health, and urban planning, making it a powerful tool for

addressing contemporary challenges in urban development, disaster preparedness, and sustainable city planning. As climate change and urbanization continue to reshape the planet, toolkits like this dataset have increasingly vital roles for understanding future climate-change- and urbanization-driven risks and impacts, further opening up new avenues for research into context-specific guidance for climate-sensitive urban planning and actionable climate adaptation strategies.

## Author contributions

L.Z. and T.C. proposed and designed the study. Y.Cheng performed the data processing, modeling and analysis, and developed the final data product. L.Z., T.C., K.O., M.D., W.L. and Y.Z. contributed ideas to the data collection, modeling, validation, and analysis. Y.Che and W.L. constructed and trained the global building height model, and contributed to building height data curation and analysis. X.L. developed the global AC penetration rate dataset. Y.Cheng and L.Z. drafted the manuscript. All authors edited and revised the manuscript.

## Competing interests

One co-author is a member of the editorial board of Earth System Science Data.

## Acknowledgements

L.Z. acknowledges the support by the U.S. National Science Foundation (CAREER Award Grant No. 2145362) and the Institute for Sustainability, Energy, and Environment at the University of Illinois Urbana-Champaign. L.Z. also acknowledges the support by the U.S. National Aeronautics and Space Administration (NASA) through the LCLUC and IST programs (Grant #80NSSC25K7322). T.C. acknowledges the supported by the U.S. Department of Energy (DOE), Office of Science, Biological and Environmental Research program through the Early Career Research Program. Pacific Northwest National Laboratory is operated for DOE by Battelle Memorial Institute under contract DE-AC05-76RL01830. Contributions from K.O. are based upon work supported by the NSF National Center for Atmospheric Research, which is a major facility sponsored by the U.S. National Science Foundation under Cooperative Agreement No. 1852977. M.D. acknowledges the support of the European Union's HORIZON Research and Innovation Actions under grant agreement No 101137851, project CARMINE (Climate-Resilient Development Pathways in Metropolitan Regions of Europe). We acknowledge the high-performance computing support provided by NSF NCAR's Computational and Information Systems Laboratory, sponsored by the U.S. National Science Foundation. We also thank Justin Braaten of the Google Earth Engine team for helping debug our initial workflow.

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
