# Peer review of "U-Surf: a global 1km spatially continuous urban surface property dataset for kilometer-scale urban-resolving Earth system modeling"

_Earth System Science Data, 2024_

## Community Comment (CC1)

This study develops a global 1km spatially continuous urban surface property dataset (U-Surf) for kilometer-scale urban-resolving Earth system modeling by leveraging the latest advances in remote sensing, machine learning, and cloud computing to provide the most relevant urban surface biophysical parameters. Compared to the default urban surface property dataset, the U-Surf dataset significantly improves the representation of urban land heterogeneity both within and across cities globally. The accuracy, uncertainties, and limitations of the U-Surf dataset are assessed and discussed. Its great value for applications is outlined as well. Overall, the manuscript is well-structured and straightforward. The developed urban surface property dataset is of great importance for urban modeling and study. I recommend the publication and just have a few comments (quite minor) for clarification.

1. How are the raw U-Surf data separated into values for the four urban density classes (e.g., TBD, HD, MD and LD, as shown in Figure 3)? Does this separation follow the locations defined by Oleson and Feddema (2020)? If it does, is the location data also provided at a 1 km resolution?

2. When the authors aggregated the 1km U-Surf data to coarser resolutions of $0.125^{\circ}$ and $1^{\circ}$, were the urban surface property parameters averaged with the weights of the fractional coverage of different 1km urban land types?

3. With the U-Surf data, the possible improvements to the urban climate simulations could be speculated in detail. For example, currently, the simulated UHI effects are overestimated in CESM2 (Liu et al., 2024). Can the new data improve this simulation?

Reference:

Liu, S., Han Y., ... & Wang, Y. (2024). More heavy precipitation in world urban regions captured through a two-way subgrid land-atmosphere coupling framework in the NCAR CESM2. Geophysical Research Letters, 51, e2024GL108747. https://doi.org/10.1029/2024GL108747.

---

## Author Comment (AC1)

**Response to reviews on *Earth System Science Data* Manuscript 2024-416 "U-Surf: a global 1km spatially continuous urban surface property dataset for kilometer-scale urban-resolving Earth system modeling"**

We thank the Editor and the Reviewers for their constructive suggestions and questions, and appreciate the opportunity to address their concerns and improve the manuscript. We address all the concerns raised by the Reviewers on a point-by-point basis. The Reviewers' original comments are indicated in blue, and our responses are indicated in black, with tracked changes in red. Please note that, for the Reviewers' convenience, all the line numbers below indicate the line numbers in the tracked-changes version of the manuscript unless otherwise stated.

**Response to Reviewers' comments:**
* * *
**Reviewer #1**
* * *
1. "This study develops a global 1km spatially continuous urban surface property dataset (U-Surf) for kilometer-scale urban-resolving Earth system modeling by leveraging the latest advances in remote sensing, machine learning, and cloud computing to provide the most relevant urban surface biophysical parameters. Compared to the default urban surface property dataset, the U-Surf dataset significantly improves the representation of urban land heterogeneity both within and across cities globally. The accuracy, uncertainties, and limitations of the U-Surf dataset are assessed and discussed. Its great value for applications is outlined as well. Overall, the manuscript is well-structured and straightforward. The developed urban surface property dataset is of great importance for urban modeling and study. I recommend the publication and just have a few comments (quite minor) for clarification."

Thank you very much. We appreciate the reviewer's acknowledgement of the significance of our study. We have addressed the reviewer's concerns in points below.

2. "How are the raw U-Surf data separated into values for the four urban density classes (e.g., TBD, HD, MD and LD, as shown in Figure 3)? Does this separation follow the locations defined by Oleson and Feddema (2020)? If it does, is the location data also provided at a 1 km resolution?"

Thank you for the good question. First, we would like to clarify that U-Surf directly provides spatially continuous UCP values without relying on any mapping from categorical urban density classes, which is why the density class data was not included in the raw U-Surf dataset. In Figure 3, we separated the raw U-Surf pixels into the four density classes just for the ease of comparison with J2010/OF2020. The reviewer is correct that this separation strictly follows the locations of the 4 density classes defined by Oleson and Feddema (2020) at 1 km resolution.

The reviewer has also raised a good suggestion of providing the location data. We have now modified the text as below to improve the clarity and provide the location data:

"U-Surf demonstrates significant improvements over the default CLMU parameters. As U-Surf directly provides spatially continuous urban surface parameters without relying on any density class or land use classification, here just for the ease of comparison and illustrative purpose, we separated raw U-Surf pixels into the four urban density classes (TBD, HD, MD, and LD) following their locations defined by J2010 and plot the distributions of the urban surface parameters in both U-Surf and J2010 data at these locations (Figure 3). The location data defined by J2010/OF2020 at 1 km resolution can be accessed at https://doi.org/10.6084/m9.figshare.28169324.v1. The overall distribution of U-Surf raw data is also shown in the figure." (Line 398-404)

3. "When the authors aggregated the 1km U-Surf data to coarser resolutions of 0.125° and 1°, were the urban surface property parameters averaged with the weights of the fractional coverage of different 1km urban land types?"

The reviewer has raised an excellent point. We greatly thank the reviewer for this insightful question, and it significantly helps improving the quality of the dataset. After carefully examining all our derived parameters in U-Surf, we agree with the reviewer that some of the property parameters should be aggregated with the weights of fractional area. In light of the reviewer's suggestion, our approach to aggregating the 1km U-Surf data to coarser resolutions (0.125° and 1°) has been refined. We have updated Figure 7 in the main text and the 1° surface data in U-Surf data version 1.1 (https://doi.org/10.5281/zenodo.14695837). In addition, we have modified the results and supplemental information as shown below.

"To demonstrate this point, we aggregated the 1km U-Surf data to coarser resolutions of 0.125° and nominal 1° (a typical resolution that ESMs are run at) to compare with J2010 side by side. More detailed information about the aggregation process can be found in Supplementary Text S1, Table S1 and Figure S27. For illustrative purposes, only the comparisons of $H/W$ are shown here." (Line 611-614)

**Supplementary Text 1: Aggregation of 1km raw U-Surf data to coarser resolutions**
 "The aggregation of urban canopy parameters (UCPs) from 1km to coarser resolutions requires careful consideration of the physical properties and conservation principles of different parameters. As shown in Table S1, we have classified UCPs into two categories, area-based conservative and non-conservative. We employed direct spatial averaging for urban percentage. For conservative parameters – roof and pervious fraction – which are inherently weighted by area, we used their urban percentages as weights to aggregate. We implemented a facet-fraction weighted averaging method for all the non-conservative parameters to ensure physically meaningful aggregation. For example, when aggregating roof or impervious canyon floor emissivity, we used the respective facet areal fractions (roof or impervious canyon floor fraction × urban percentage) with respect to the 1 km grid as their weights. This way ensures that the contribution of each parameter to the coarser resolution is proportional to its actual surface coverage.

The aggregation of canyon height-to-width ratio ($H/W$) is slightly more complex as it is derived from multiple primary parameters. We evaluated two potential aggregation methods: 'aggregating first' and 'aggregating after' (Li et al., 2024), both using urban density (urban percentage × roof fraction) as weights. The former is to aggregate the primary input parameters (e.g., building height, roof fraction) to the target resolution before calculating $H/W$. The latter calculates $H/W$ at the

original 1km resolution before spatial aggregation. Our analysis revealed that the 'aggregating after' method generally produces slightly higher values and preserves more spatial variation compared to the 'aggregating first' method (Figure S27). In addition, the 'aggregating after' method better maintains the non-linear relationships between input and output parameters and hence preserves local canyon characteristics during the upscaling process. This choice aligns with the recommendations from previous studies (e.g., Dai et al., 2019; Shangguan et al., 2014) and helps prevent the smoothing of local variations in the 'aggregating first' method. Therefore, in the published dataset with this study we choose the 'aggregating after' method to aggregate $H/W$ to coarser resolutions (0.125° and 1°)."

**Table S1. Conservativeness of urban surface property parameters under spatial aggregation**

| Category | Parameter | Facet type | | | |
|---|---|---|---|---|---|
| | | Roof | Impervious canyon floor | Pervious canyon floor | Wall |
| **Radiative** | Emissivity | N | N | N | N |
| | Albedo | N | N | N | N |
| **Morphological** | Fraction* | Y | - | Y | - |
| | Building height | N | - | - | - |
| | Canyon height-to-width ratio | N | - | - | - |
| **Thermal** | Thickness | N | - | - | N |
| | Volumetric heat capacity | N | N | - | N |
| | Thermal conductivity | N | N | - | N |

Y: conservative parameters; N: non-conservative parameters; -: not applicable.
* Urban percentage is another fractional (conservative) parameter.

[Figure]

**Figure S27. Comparison between 'aggregating fist' and 'aggregating after' method for aggregating** $H/W$ **from 1km to 1° in selected regions.**

4. "With the U-Surf data, the possible improvements to the urban climate simulations could be speculated in detail. For example, currently, the simulated UHI effects are overestimated in CESM2 (Liu et al., 2024). Can the new data improve this simulation?"

We thank the reviewer for this insightful question regarding the potential impact of U-Surf data on urban climate simulations. To address the reviewer's question, we have run several sets of preliminary land-only simulations spanning from 2010 to 2014 forced by bias-corrected ERA5 (Cucchi et al., 2020) with CESM2. Below we calculated (i) the canopy UHI as the difference between urban and rural near-surface air temperature and (ii) the surface UHI as the difference between urban and rural land surface temperature for each grid.

We have added more details in broader implication (section 4) and a new supplementary Figure S28 as below.

"By integrating the latest global data sources, U-Surf provides global continuity and local granularity in urban surface representation. This enhanced representation shows promise in correcting systematic biases in current models and improving their modeling accuracy and predictability. For example, a recent study finds that the simulated urban heat island (UHI) effects tend to be overestimated in CESM2 (Liu et al., 2024). To test the effects of U-Surf, we have run two preliminary land-only CESM2 simulations (0.9375°×1.25°) spanning from 2010 to 2014 with the default urban surface data and U-Surf, both forced by bias-corrected ERA5 (Cucchi et al., 2020). We find that this overestimation is largely reduced by an average of 0.176K in annual canopy UHI (CUHI) over China, due to the widespread cooling trend in urban near-surface air temperatures (Figure S28). This improvement aligns with Liu et al., (2024)'s findings that CESM2

overestimates CUHI in China by +0.127°C. Moreover, the remote-sensing-based methodology offers a unique capability to track the quantitative evolution of urban canopy parameters (UCPs) over time, a level of detail that is difficult to extract from traditional classification methods." (Line 739-748)

[Figure]

**Figure S28. Spatial distribution of 5-year average canopy urban heat island (UHI) intensity [K] in China.** The panels show (a) annual, (b) summer (JJA), and (c) winter (DJF) averages simulated by CESM2 using default urban canopy parameters (UCPs) (top row), U-Surf parameters (middle row), and their difference (U-Surf minus default, bottom row). Negative values in the difference plots indicate weaker UHI intensity with U-Surf parameters.

We have observed similar pattern of reduced CUHI in a higher-resolution simulation (1/32°) over CONUS (Figure R1). More significant reduction of surface UHI (SUHI) have been shown from the same simulation (Figure R2), which can be primarily attributed to increased emissivity values compared to J2010/OF2020. This finding aligns with Chakraborty et al., (2021), which demonstrated stronger sensitivity of SUHI intensity to surface emissivity.

In summary, integrating U-Surf parameters into CESM simulations could indeed reduce the overestimation of both SUHI and CUHI. A comprehensive assessment of U-Surf's impacts would require coupled simulations to account for land-atmosphere interactions.

[Figure]

**Figure R1. Same as S29 but for 1/32° simulation results over contiguous U.S.**

[Figure]

**Figure R2. Same as R1 but for surface UHI intensity.**

**Response to Reviewers' comments:**
* * *
**Reviewer #2**
* * *
1. "Dear authors,
High-quality urban surface property dataset is vital for high-resolution urban climate modeling, this study use a series of mature methods to generate a global spatially continuous dataset (named as: U-Surf) from multisourced remote sensing observations and products, which contains radiative, morphological, and thermal properties. Overall, the methodological framework is complete and the generated products are valuable."

Thank you very much for acknowledging the importance of our study and the U-Surf product. We have addressed the reviewer's concern in points below.

2. The method descriptions should be totally strengthened.
2.1 For example, in Section 2.2.1, I don't know how to use the ASTER and Sentinel-2 imagery (the yearly-composited imagery or all available imagery) to calculate the single or time-series emissivity and albedo.

Thank you for raising these important points. We have added more details in Section 2.2.1 as shown below and updated Table 1 accordingly:

"This facet-segmented image was then applied to the Advanced Spaceborne Thermal Emission and Reflection (ASTER) Global Emissivity Dataset 100 m V003 product (hereafter referred as ASTER GEDv3; Hulley et al., 2015) and the Sentinel-2 land surface albedo data (Lin et al., 2022) to extract the emissivity and albedo of building roof, and impervious and pervious ground. The static emissivity imagery is composited from clear-sky (cloud free) pixels for all available ASTER data from 2000 to 2008 (Hulley et al., 2015) to represent the emissivity climatology over this period. We use a linear spectral-to-broadband algorithm (Malakar et al., 2018) to estimate the broadband emissivity from ASTER GEDv3 bands (Eq. 1):" (Line 212-218)

"For albedo, we used a 10 m land surface blue-sky albedo product retrieved from Sentinel-2 which covers nearly 2,300 major cities across the globe (Lin et al., 2022). For the rest of the global urban areas, we applied the narrow-to-broadband conversion method (Bonafoni and Sekertekin, 2020) to estimate the 10m-resolution albedo based on Sentinel-2 surface reflectance (Eq. 2). Both the blue-sky albedo product and the narrow-to-broadband calculated albedo are derived using the Sentinel-2 imageries composited from 2019 to 2021. The blue-sky albedo product only includes cloud free images. For the narrow-to-broadband algorithm, we use the Cloud Score+ (CS+) dataset (Pasquarella et al., 2023) to mask out the cloud-contaminated pixels, where pixels with a CS+ quality assessment score below 0.8 were excluded." (Line 228-235)

2.2 As for the Eq. (4), how to determine the parameter of wf, which follows the normal distribution? Equal distribution?

In Eq. (4), $\alpha_f^{1km} = \frac{\sum_{i=0}^{9999} w_f^i \cdot \alpha_i^{10m}}{\sum_{i=0}^{9999} w_f^i}$, $w_f^i$ is the areal fractions of a certain facet with respect to each 10m grid cell. The subscript $f$ stands for each individual facet (such as roof, wall, etc.). These fractions are calculated by dividing the area of each facet by the area of the grid at each grid. Therefore, they don't necessarily follow Equal or normal distribution. Most of these numbers are concentrated in the low end as very densely built cities are scarce worldwide. We have revised the text accordingly:

"where $\varepsilon_f^{1km}$ and $\alpha_f^{1km}$ is 1 km emissivity and albedo, respectively, for a certain facet (roof/impervious canyon floor/pervious canyon floor), $w_f^i$ is the area fractions of a certain facet within each 100m or 10m grid cell derived from the 10m segmented imagery. The subscript $f$ stands for each individual facet. For example, when calculating roof emissivity and albedo, $w_f^i$ is the roof fraction within each 100m and 10m grid cell, respectively." (Line 249-252)

2.3 As for the albedo and emissivity model in Eq. (2-3), how to consider their uncertainty? How do these models perform on a global scale?

The stepwise regression approach (Eq. (1); Malakar et al., (2018)) was originally adapted from Ogawa et al., (2008). The derived ASTER broadband emissivity was validated against 305 samples from ASTER spectral library covering the wavelength ranging from 2-15$\mu m$, yielding the $R^2$ of 0.913 and RMSE of 0.011. The spectra covered different land covers including rocks, soils, vegetation and water with the spatial focus in California.

The model has been used and validated in multiple studies (Chakraborty et al., 2021; Ru et al., 2023). To address the reviewer's question, we have now added this information in the text and updated Table 3:

"ASTER GEDv3 (Hulley et al., 2015) was compared with MODIS Collection 4 & 5 Emissivity and validated against lab measurements at four large sand dune fields, yielding a relatively low average RMSE of 0.077. The broadband emissivity regression algorithm (Eq. 1) was validated against ASTER spectral library covering the wavelength ranging from 2-15μm, yielding the R2 of 0.913 and RMSE of 0.011 (Malakar et al., 2018; Ogawa et al., 2008)." (Line 692-696)

As we have discussed in Section 3.4, according to Bonafoni and Sekertekin, (2020), the narrow-to-broadband albedo model (Eq. 2) has demonstrated credibility on the global scale through two validation efforts. The first validation against six stations of SURFRAD generated an average R of 0.77 and RMSE of 0.023. The second validation across 18 urban sites in Perugia, Italy yielded an average R of 0.98 and RMSE of 0.021. The validation sites are carefully chosen to include different landcover types (cropland, grassland/sparse grassland, natural vegetation, urban) and background climate (humid continental, cold semi-arid, humid subtropical, mediterranean), the consistent performance across the heterogeneous site can demonstrate the model's robustness.

For the detailed uncertainty analysis and error propagation, please refer to our response to comment #3 below.

2.4 Similarly, some details should be added in the Section 2.2.2 and 2.2.3.

In response to the reviewer's comments, we have revised the text to improve the clarity as below.

"The 3D-GloBFP is a global building height data at a building footprint scale recently developed by leveraging a combination of Synthetic Aperture Radar (SAR), optical imagery, terrain, population, nighttime light data primarily covering 2014 to 2023, and XGBoost machine learning approach. We aggregated the vector-level height to 1 km grids using area-weighted averages. The second global building height data (Li et al., 2022) is a raster map at 1 km spatial resolution that also utilizes radar and optical satellite imagery, along with additional geographical information circa 2015." (Line 285-290)

"Canyon height-to-width ratio ($H/W$; i.e., the ratio of building height to canyon width) is another critical morphological parameter that is widely used in most UCMs including CLMU. It is a proxy parameter that implies the structural layout and compactness of the built area. Unlike other parameters that can be directly measured by satellite data, $H/W$ needs to be derived on the basis of model geometry and assumptions. Consistent with the single-layer urban canyon geometry in UCMs, the $H/W$ in this study is estimated using the 2D infinite street canyon model with two recommended primary parameters, building fraction (or plan area density; $\lambda_p$) and wall surface density ($\lambda_w$) (Masson et al., 2020):" (Line 295-301)

"This table includes thickness, thermal conductivity and volumetric heat capacity of up to 10 layers for common types of roofs, walls, roads (layers with identical materials are allowed) (Oleson and Feddema, 2020). As these thermal parameters are provided in a look-up table instead of a geospatially explicit format, we need to map the table values to each 1 km grid in U-Surf. In order to do this, we classified 1 km U-Surf urban grids into four nominal density classes: TBD (0.016% of the pixels), HD (3.83%), MD (41.98%), LD (54.17 %) (Figure S2) based on the percentiles of canyon height-to-width ratio defined in J2010. We then applied the corresponding thermal parameters from the lookup table to each class to ensure it covers all possible materials used in 33 regions (Figures S18-S25). Although this is likely the most feasible approach for providing an ESM-compatible global building thermal property dataset at present, we acknowledge its limitation of relying somewhat on coarse-grained regional and density-class values. Once more detailed, spatially explicit global datasets – such as those on building materials or thermal properties – become available, we can readily incorporate their thermal parameters into future releases of U-Surf."(Line 327-337)

3. My major concern is the quality of the U-Surf. Although the authors emphasized that "validating urban surface parameters on the global scale is extremely challenging primarily due to the lack of globally consistent measurement networks", I don't think the Table 3 can support the accuracy analysis of the U-Surf. My concerns come from that 1) the synthesized data products only part of the parameters of the retrieved models in Section 2.2, i.e., how to quantify the transformed error of synthesized data in the retrieved models; 2) as a user of the U-Surf, I also want to know its absolute accuracy not that its better than the previous data. I hope that the authors can strength the accuracy assessment.

We thank the reviewer for pointing this out and for the good suggestion. We agree with the reviewer that a more comprehensive and explicit accuracy/uncertainty assessment would strengthen the U-Surf product. To address the reviewer's concern, we have improved the accuracy/uncertainty assessment section thoroughly by explicitly quantifying the error propagation for all the parameters in U-Surf.

Specifically, for parameters derived from multiple data sources, we use Monte Carlo (MC) simulation approach to quantify how uncertainties propagate through our calculations. We used the documented uncertainties of input datasets (updated Table 3) as probability distributions in the MC simulations to quantify the expected error ranges in our final estimated parameters. These simulation results are now presented in the new Table 4 in the revised manuscript.

1) Uncertainty analysis
We first selected 10 countries/regions in total that cover every continent (except for Antarctica) and diverse Koppen climate zone globally (Supplementary Table S2). Then, for each country we randomly sampled 1,000 pixels, forming a total sample size of 10,000 to conduct MC simulations.

**Table S2.** Selected countries/regions to demonstrate the uncertainty propagation.

| Continent | North America | | South America | | Europe | | Asia | | Oceania | Africa |
|---|---|---|---|---|---|---|---|---|---|---|
| **Country** | United States | Mexico | Argentina | Bolivia | France | Poland | China | Malaysia | Australia | Nigeria |
| **Koppen climate zone\*** | - | BSh | Cfa | Aw | Cfb | Dfb | - | Af | Bwh | Aw |
| | Varying | Hot semi-arid | Humid subtropical | Tropical savanna | Temperate oceanic | Warm summer continental | Varying | Tropical rainforest | Hot desert | Topical savanna |

\* The table only shows the predominant koppen climate zone if any.

For the following sections we just use roof albedo as an example to illustrate how we quantified the uncertainty propagation during data synthesize and spatial aggregation through MC simulation approach. Similar procedure was repeated for all U-Surf radiative and morphological parameters. The results have been reported in the revised manuscript.

The derivation of roof albedo at 1-km resolution includes uncertainties from three components: building footprints detection, 10-m Sentinel-2 blue sky albedo product, narrow-to-broadband (NTB) conversion algorithm. The uncertainty from building footprint detection can be quantified using the false positive rate, which is approximately 1% in North America according to Table 3, i.e. the error is $\sigma_{roof} = 0.01$. The RMSE of narrow-to-broadband algorithm is $\sigma_{a2} = 0.023$. Regarding the uncertainty of Sentinel-2 albedo product, we can estimate it by examining the RMSE of black-sky albedo $\alpha_{black}$ and white-sky albedo $\alpha_{white}$. According to Lin et al., (2022), $\sigma_{black} = 0.0185$ and $\sigma_{white} = 0.0205$, reported as the average of the unevenly and uniformly distributed urban area. We calculated the blue sky albedo $\alpha_{blue}$ as $(1 - D)\alpha_{black} + D\alpha_{white}$, where $D$ is the diffuse skylight ratio, with the common of value of 0.3 using the BaRAD2019 dataset from Chakraborty and Lee, (2021). Thus, $\sigma_{black} = \sqrt{(1 - D)^2 \alpha_{black}^2 + D^2 \alpha_{white}^2} = 0.0154$.

Theoretically, if we assume the uncertainties from different data products are independent, we can add the uncertainty from each data sources as a normally distributed perturbation to test the sensitivity of roof albedo to these uncertainties. For most of our input data, only RMSE value was available. Since $RMSE^2 = bias^2 + variance$, we adopted a conservative approach by using RMSE as the upper limit of potential bias, thereby maximizing our uncertainty estimation. We ran the Monte Carlo simulation by repeating aforementioned process by 1,000 times and taking the standard error across all simulations to analyze the propagation of uncertainties. Since we only applied the NTB algorithm where Sentinel-2 blue sky albedo product is unavailable, we can easily combine the uncertainties generated from two procedures together to calculate the final uncertainty.

The average 95% confidence interval (i.e., $z = 1.96$) of roof albedo across selected countries is approximately 0.003-0.017 at 1km resolution (Table 4). This demonstrates a reduction in uncertainty compared to the input data uncertainties at 10m resolution, primarily due to the spatial aggregation process from 10m to 1km resolution.

**Table 4. Estimated 95% confidence intervals (±) by Monte Carlo simulations across all regions.**

| Continent | Country | Radiative | | | | | | | | Morphological | | | | |
| | | Emissivity | | | | Albedo | | | | | | | | |
| | | Roof | Impervious canyon floor | Pervious canyon floor | Wall* | Roof | Impervious canyon floor | Pervious canyon floor | Wall* | Urban percentage | Roof fraction | Pervious fraction | Canyon height to width ratio | Building height (m) |
|---|---|---|---|---|---|---|---|---|---|---|---|---|---|---|
| North America | United States | 0.0443 | 0.0369 | 0.0181 | 0.0443 | 0.0086 | 0.0067 | 0.0020 | 0.0086 | 0.0029 | 0.0019 | 0.0056 | 0.0751 | 3.8092 |
| | Mexico | 0.0316 | 0.0326 | 0.0137 | 0.0316 | 0.0043 | 0.0058 | 0.0009 | 0.0043 | 0.0039 | 0.0029 | 0.0060 | 0.1702 | 5.0254 |
| South America | Argentina | 0.0342 | 0.0289 | 0.0169 | 0.0342 | 0.0034 | 0.0044 | 0.0012 | 0.0034 | 0.0048 | 0.0024 | 0.0059 | 0.2130 | 5.0589 |
| | Bolivia | 0.0477 | 0.0443 | 0.0133 | 0.0477 | 0.0096 | 0.0099 | 0.0010 | 0.0096 | 0.0033 | 0.0029 | 0.0076 | 0.1307 | 5.2250 |
| Europe | France | 0.0386 | 0.0479 | 0.0149 | 0.0386 | 0.0047 | 0.0101 | 0.0010 | 0.0047 | 0.0031 | 0.0045 | 0.0073 | 0.1592 | 4.4660 |
| | Poland | 0.0381 | 0.0450 | 0.0159 | 0.0381 | 0.0045 | 0.0082 | 0.0011 | 0.0045 | 0.0031 | 0.0046 | 0.0085 | 0.1585 | 4.5200 |
| Asia | China | 0.0428 | 0.0394 | 0.0164 | 0.0428 | 0.0169 | 0.0056 | 0.0012 | 0.0169 | 0.0047 | 0.0061 | 0.0104 | 0.2156 | 8.2679 |
| | Malaysia | 0.0359 | 0.0432 | 0.0135 | 0.0359 | 0.0041 | 0.0082 | 0.0007 | 0.0041 | 0.0034 | 0.0024 | 0.0049 | 0.1058 | 2.8700 |
| Oceania | Australia | 0.0542 | 0.0616 | 0.0126 | 0.0542 | 0.0080 | 0.0129 | 0.0007 | 0.0080 | 0.0023 | 0.0026 | 0.0074 | 0.0681 | 3.0307 |
| Africa | Nigeria | 0.0432 | 0.0560 | 0.0202 | 0.0432 | 0.0069 | 0.0118 | 0.0019 | 0.0069 | 0.0023 | 0.0087 | 0.0107 | 0.1052 | 2.5947 |
| Average | | 0.0411 | 0.0436 | 0.0156 | 0.0411 | 0.0071 | 0.0084 | 0.0012 | 0.0071 | 0.0034 | 0.0039 | 0.0074 | 0.1401 | 4.4868 |

*Wall radiative parameters were processed by directly utilizing the roof parameters, resulting in the same uncertainty range.

**2) Absolute accuracy**

To validate our morphological parameters thematically, we compared U-Surf data against two recently available, observation-based datasets, Urban-PLUMBER and WSF-3D. This comparison, as visualized in Figure 8 and S5 (now moved to Figure 9 in the revised manuscript), shows generally strong agreement, especially for roof fraction (MAE=0.076, 0.081 for two reference datasets, respectively). We have added the data and bias as new supplementary Table S3 and S4. While direct validation of radiative parameters remains challenging due to the lack of comprehensive ground-truth datasets, our uncertainty analysis above demonstrates that the error propagation through our data synthesis and processing remains considerably small (Table 4).

**Table S3. Thematic validation results at 21 Urban-PLUMBER sites.**

| Site | City | Country | Roof fraction MAE | Pervious fraction MAE | Building height MAE (m) | Canyon height-to-width ratio MAE |
|---|---|---|---|---|---|---|
| AU-Preston | Melbourne | Australia | 0.110 | 0.370 | 0.929 | 0.124 |
| AU-Surreyhills | Melbroune | Australia | 0.061 | 0.015 | 4.407 | 0.204 |
| CA-Sunset | Vancouver | Canada | 0.030 | 0.111 | 1.664 | 0.159 |
| FI-Kumpula | Heksinki | Finland | 0.005 | 0.005 | 0.887 | 0.013 |
| FI-Torni | Helsinki | Finland | 0.165 | 0.112 | 4.220 | 0.582 |
| FR-Capitole | Toulouse | France | 0.014 | 0.046 | 2.224 | 0.446 |
| GR-HECKOR | Crete | Greece | 0.025 | 0.023 | 1.484 | 0.855 |
| JP-Yoyogi | Tokyo | Japan | 0.065 | 0.180 | 16.354 | 0.816 |
| KR-Jungnang | Seoul | South Korea | 0.397 | 0.014 | 2.900 | 0.441 |
| KR-Ochang | Ochang | South Korea | 0.022 | 0.271 | 12.600 | 0.056 |
| MX-Escandon | Mexico City | Mexico | 0.042 | 0.180 | 2.562 | 1.019 |
| NL-Amsterdam | Amsterdam | Netherlands | 0.111 | 0.194 | 3.780 | 0.240 |
| PL-Lipowa | Lodz | Poland | 0.052 | 0.101 | 21.590 | 0.316 |
| PL-Narutowicza | Lodz | Poland | 0.059 | 0.078 | 1.008 | 0.358 |
| SG-TelokKurau06 | - | Singapore | 0.160 | 0.118 | 15.737 | 1.493 |
| UK-KingsCollege | London | United Kingdom | 0.120 | 0.016 | 1.652 | 0.696 |
| UK-Swindon | Swindon | United Kingdom | 0.049 | 0.054 | 2.446 | 0.010 |
| US-Baltimore | Baltimore | United States | 0.044 | 0.002 | 7.239 | 0.082 |
| US-Minneapolis1 | Minnesota | United States | 0.050 | 0.047 | 6.436 | 0.057 |
| US-Minneapolis2 | Minnesota | United States | 0.060 | 0.114 | 6.436 | 0.057 |
| US-WestPhoenix | Arizona | United States | 0.050 | 0.561 | 7.716 | 0.113 |
| Average | | | 0.081 | 0.124 | 5.918 | 0.387 |

**Table S4. Thematic validation results at 17 WSF-3D sites.**

| Site | Country | Roof fraction MAE | Roof fraction RMSE | Building height MAE (m) | Building height RMSE (m) |
|---|---|---|---|---|---|
| Almaty | Kazakhstan | 0.054 | 0.070 | 8.150 | 10.586 |
| Amsterdam | Netherlands | 0.070 | 0.095 | 5.098 | 7.078 |
| Bavaria | Germany | 0.065 | 0.084 | 4.217 | 5.955 |
| Cartagena | Colombia | 0.088 | 0.116 | 8.551 | 11.996 |
| Dongying | China | 0.111 | 0.130 | 7.627 | 9.805 |
| Gyeonggi | South Korea | 0.067 | 0.083 | 6.029 | 8.378 |
| Indianapolis | United States | 0.045 | 0.067 | 5.000 | 7.264 |
| Kigali | Rwanda | 0.062 | 0.084 | 6.475 | 9.823 |
| Lipa | Philippines | 0.094 | 0.104 | 5.274 | 6.542 |
| Munich | Germany | 0.053 | 0.069 | 5.587 | 7.626 |
| Nairobi | Kenya | 0.058 | 0.082 | 12.445 | 17.714 |
| NewYork | United States | 0.076 | 0.105 | 11.653 | 14.889 |
| Niamey | Niger | 0.112 | 0.133 | 10.636 | 14.473 |
| Seoul | South Korea | 0.113 | 0.124 | 11.210 | 13.386 |
| Tanauan | Philippines | 0.092 | 0.104 | 4.881 | 5.675 |
| Vienna | Austria | 0.063 | 0.082 | 6.614 | 8.872 |
| Washington | United States | 0.072 | 0.094 | 7.132 | 8.276 |
| Average | | 0.076 | 0.096 | 7.446 | 9.902 |

To address the reviewer's concern and in light of the reviewer's suggestion, we have improved the Accuracy assessment and uncertainty section (Section 3.4) by adding discussions on the uncertainty quantification using MC simulations (new Supplementary Text S2) and accuracy evaluation against observations. We have also revised the high-level summary in Section 2.4 accordingly:

[revised manuscript text omitted]

Supplementary Text S2: Uncertainty propagation in data synthesis and processing
"In our uncertainty assessment, we employed Monte Carlo simulation approach that assume the uncertainties from different data products are independent. For each simulation, we introduced normally distributed perturbations based on the documented uncertainties of individual data sources (Table 3) to evaluate how these variations affect our 1km output parameters. Most input datasets provided only RMSE values as their uncertainty metric, thus we adopted a conservative approach by approximating the standard deviation with RMSE ($RMSE^2 = bias^2 + variance$, thus $variance \leq RMSE$) in these cases, thereby ensuring our uncertainty estimates remain conservative.

For most input datasets, we could directly obtain uncertainty values from original literature or calculate them through simple averaging. However, the uncertainty estimation for the Sentinel-2 blue-sky albedo product required additional steps. We estimate the uncertainty of the Sentinel-2 blue-sky albedo by examining the RMSE of black-sky albedo $\alpha_{black}$ and white-sky albedo $\alpha_{white}$. According to Lin et al. (2022), the average uncertainty of the unevenly and uniformly distributed urban areas gives $\sigma_{black} = 0.0185$ and $\sigma_{white} = 0.0205$, respectively. We then calculated the blue-sky albedo $\alpha_{blue}$ as $(1 - D)\alpha_{black} + D\alpha_{white}$, where $D$ is the diffuse skylight ratio and is assigned the commonly used value of 0.3 here based on the BaRAD2019 dataset from Chakraborty and Lee, (2021). Thus, $\sigma_{black} = \sqrt{(1 - D)^2 \alpha_{black}{}^2 + D^2 \alpha_{white}{}^2} = 0.0154$."

4. The comparisons and analysis in the Figure 4 and 5 are interesting, meanwhile, I hope that authors can add some quantitative statistics. For example, in the line of 453-454, the author stated "the Global South (Latin America, Africa, and parts of Asia) generally shows lower values for these parameters and higher pervious surface fractions". If the further statistics and analysis can be added, it may be interesting.

Thank you for pointing this out. We have added more quantitative statistics and discussions in the main text as below:

"In the Global North, particularly in Europe and United States, urban areas typically exhibit higher building density (roof fraction × urban percentage), greater average building height, and higher average canyon height-to-width ratio. These characteristics are indicative of more developed urban form and well-established infrastructure, often driven by the need to accommodate growing

populations in limited spaces. For instance, metropolitan centers (e.g. Manhattan, New York City, USA; Quartiers 1-4, Paris, France) in these areas frequently exceed 30-40% roof coverage, with average building heights surpassing 30 meters. In contrast, regions in the Global South (e.g., Latin America, Africa, and Central Asia) generally exhibit lower values for these parameters. For example, building density in these regions are 38.59%, 46.46%, 88.71% lower, respectively, than in the United States. Similarly, their median building height is 11.94%, 31.65%, 12.75% lower, respectively, than in Europe. Consequently, their median canyon height-to-width ratios are 29.88%, 37.18%, 23.99% lower, respectively, than those in Europe. However, this trend is rapidly changing in emerging economies, including India and Brazil, where cities are experiencing swift urban growth. For instance, rapidly urbanizing places such as Delhi, India and Sao Paulo, Brazil have demonstrated tall and densely built environments, where Delhi has a roof fraction of 31.02% and building heights of 12.63m, while Sao Paulo has a roof fraction of 49.42% and building heights of 13.87m, all of which exceed the 75th percentile in the global distribution (Figure 3c). Additionally, regions such as East Asia exhibit urbanization patterns that are more akin to those in North America and Europe, characterized by high roof fractions (Figure S4a) and significant vertical development. For example, many cities in Eastern China have exhibited city-wide average roof fractions above 14% and building heights exceeding 13m, reflecting rapid industrialization and economic growth that have rapidly transformed the urban landscape over the past few decades (Cai et al., 2022)." (Line 516-551)

5. The descriptions about the TBD, HD, MD and LD should be strengthen, which has been mentioned several times in the results section.

We thank the reviewer for the suggestion. First, we would like to clarify that U-Surf directly provides spatially continuous UCP values and therefore does not have categorical density classes. We use the four urban density classes (TBD, HD, MD, and LD) defined by J2010 in our results and discussion sections are just for two reasons: (i) for the ease of direct comparison with J2010/OF2020 data and (ii) to leverage the look-up table provided by J2010 for certain thermal properties. These four density classes were classified mainly based on their morphological and population density characteristics in J2010. Their typical ranges (valid for circa-2000) can be found in Jackson et al. (2010) and are reproduced here below in Table R1.

**Table R1. Characteristics of TBD, HD and MD per definition of Jackson et al. (2010).**

| Urban Class | H/W | Building Heights (m) | Pervious Fraction (%) | Population Density (km$^2$) | Typical Building Types |
|---|---|---|---|---|---|
| Tall Building District (TBD) | 4.6 | 40-200+ | 5-15 | 14,000 – 134,000+ | Skyscrapers |
| High Density (HD) Residential/ Commerical/ Industrial | 1.6 | 17-45 | 15-30 | 5,000 – 80,000+ | Tall apartments, office bldgs, industry |
| Medium Density (MD) Residential | 0.7 | 8-17 | 20-60 | 1,000 – 7,000 | 1-3 story apartment bldgs, row houses |

We have also added more information and discussion in the revised manuscript about how we separated raw U-Surf data to the four classes following their locations defined in J2010 in our response to Reviewer #1 Comment #2, and how we leveraged the four density classes to inform

our thermal parameter assignments in our response to Comment #2.4. Please refer to our responses to these comments above.

In addition, we have revised the following text to improve the clarity as below:

"J2010 clusters the global urban areas into 33 distinct regions sharing similar climates, socio-economic characteristics, and architectural practices (Figure S1), with properties defined within each region for up to four urban density classes: low density (LD), medium density (MD), high density (HD), and tall building district (TBD). These density classes are classified based on morphological features (including building height, pervious areal fraction, canyon height-to-width ratio, and typical building type) and population density. The dataset then prescribes uniform surface properties to each density type within a region." (Line 89-94)

6. In summary, the U-Surf dataset provides important support in urban climate modeling, and shows great advantages over the previous dataset (such as: CLMU and J2010), which meets the high-quality of the ESSD journal. I hope that above comments will help to improve the quality of this article.

Thank you very much. We appreciate the review's acknowledgement of the importance of U-Surf dataset and the helpful comments and questions.